# Remodeling of lumbar motor circuitry remote to a thoracic spinal cord injury promotes locomotor recovery

Ying Wang[1,2,3], Wei Wu[1,2], Xiangbing Wu[1,2], Yan Sun[1,2,4], Yi P Zhang[5], Ling-Xiao Deng[1,2], Melissa Jane Walker[1,2], Wenrui Qu[1,2], Chen Chen[1,2,6], Nai-Kui Liu[1,2], Qi Han[1,2], Heqiao Dai[1,2], Lisa BE Shields[5], Christopher B Shields[5], Dale R Sengelaub[7], Kathryn J Jones[8], George M Smith[9], Xiao-Ming Xu[1,2,8]*

[1]Spinal Cord and Brain Injury Research Group, Stark Neurosciences Research Institute, Indiana University School of Medicine, Indianapolis, United States; [2]Department of Neurological Surgery, Indiana University School of Medicine, Indianapolis, United States; [3]Neural Tissue Engineering Research Institute, Mudanjiang College of Medicine, Mudanjiang, China; [4]Department of Anatomy, Histology and Embryology, School of Basic Medical Sciences, Fudan University, Shanghai, China; [5]Norton Neuroscience Institute, Norton Healthcare, Louisville, United States; [6]Program in Medical Neuroscience, Paul and Carole Stark Neurosciences Research Institute, Indiana University School of Medicine, Indiana, United States; [7]Program in Neuroscience, Department of Psychological and Brain Sciences, Indiana University, Bloomington, United States; [8]Department of Anatomy and Cell Biology, Indiana University School of Medicine, Indianapolis, United States; [9]Shriners Hospitals Pediatric Research Center, Lewis Katz School of Medicine, Temple University, Philadelphia, United States

*For correspondence:
xu26@iupui.edu

Competing interests: The authors declare that no competing interests exist.

**Abstract** Retrogradely-transported neurotrophin signaling plays an important role in regulating neural circuit specificity. Here we investigated whether targeted delivery of neurotrophin-3 (NT-3) to lumbar motoneurons (MNs) caudal to a thoracic (T10) contusive spinal cord injury (SCI) could modulate dendritic patterning and synapse formation of the lumbar MNs. *In vitro*, Adeno-associated virus serotype two overexpressing NT-3 (AAV-NT-3) induced NT-3 expression and neurite outgrowth in cultured spinal cord neurons. *In vivo*, targeted delivery of AAV-NT-3 *into* transiently demyelinated adult mouse sciatic nerves led to the retrograde transportation of NT-3 to the lumbar MNs, significantly attenuating SCI-induced lumbar MN dendritic atrophy. NT-3 enhanced sprouting and synaptic formation of descending serotonergic, dopaminergic, and propriospinal axons on lumbar MNs, parallel to improved behavioral recovery. Thus, retrogradely transported NT-3 stimulated remodeling of lumbar neural circuitry and synaptic connectivity remote to a thoracic SCI, supporting a role for retrograde transport of NT-3 as a potential therapeutic strategy for SCI.
DOI: https://doi.org/10.7554/eLife.39016.001

## Introduction

Circuitry within the lumbar spinal cord integrates afferent information from descending and segmental sources to coordinate the patterned firing of motor neuron pools (*Cazalets et al., 1995*; *Cazalets et al., 1996*). This command structure breaks down after spinal cord injury (SCI).

Approaches aimed at reactivating the lumbar neural circuit may offer attractive strategies to restore locomotion after SCI (*Beaumont et al., 2004*; *Boulenguez and Vinay, 2009*).

Caudal to an SCI, denervation of motoneurons (MNs) can result in dendritic reorganization or atrophy (*Gazula et al., 2004*). We have previously reported that lumbar spinal MNs caudal to a thoracic contusive SCI underwent a marked reduction in their dendritic arbor (*Byers et al., 2012*; *Liu et al., 2014a*). Dendritic branching patterns and distribution determine important functional properties in MNs (*Cameron and Núñez-Abades, 2000*), and thus preventing dendritic atrophy could be an important therapeutic target for restoring neural circuitry and functional recovery after SCI.

Descending spinal tracts are critically involved in specific functions, such as fine locomotor control, neuromodulation or initiation of rhythmic movements (*Deumens et al., 2005*; *Jordan et al., 2008*). Contusive SCI can induce anatomically incomplete lesions whereas axons surrounding the lesion sites remain intact. These axons include descending supraspinal and propriospinal axons that project to spinal cord segments below the lesion. Establishing reliable strategies to enhance collateral growth and synaptogenesis from these spared descending axons to lumbar MNs, the common final pathway for hindlimb locomotion, is a feasible goal with great potential.

Neurotrophic factors can support the sprouting of descending axons into injury-induced denervated areas to form new synapses (*da Silva and Wang, 2011*; *Park and Poo, 2013*). Neurotrophin-3 (NT-3), a member of the neurotrophin family of proteins (*Barbacid, 1994*; *Snider, 1994*), is an attractive candidate for stimulating MN dendritic plasticity because it is a survival factor for MNs and is expressed by MNs, one of the main targets of descending axons of the spinal cord (*Henderson et al., 1993*; *Buck et al., 2000*). In addition, a previous study has demonstrated that delivery of NT-3 via retrograde transport of AAV from triceps to cervical motoneurons led to reduced functional loss and anatomical reorganization of the CST following a dorsal lesion at C4-5 (*Fortun et al., 2009*).

Here, we examined whether focal injections of adeno-associated virus serotype two overexpressing NT-3 (AAV-NT-3) into the sciatic nerves would result in retrograde transport of NT-3 to lumbar MNs. We hypothesized that the virus transduced MNs would release the retrogradely transported NT-3 resulting in an elevation of the local NT-3 level, promoting dendritic remodeling, stimulating sprouting and synaptogenesis of innervating descending axons, and attenuating targeted muscle atrophy. We also hypothesized that remodeling of the lumbar MN neural circuit would result in improved physiological and behavioral recoveries.

## Results

### AAV-NT-3 gene transfer enhanced NT-3 expression and neurite outgrowth of spinal cord neurons *in vitro*

We first determined the efficiency of AAV infection in spinal cord neurons *in vitro* (*Figure 1A*). We found that AAV-GFP infection in spinal cord neurons was highly efficient (infection rate: 87.4 ± 6.34%; *Figure 1A and B*). Next, we determined the expression of NT-3 in the AAV-NT-3 transfected spinal cord neurons using immunofluorescence staining and an enzyme-linked immunosorbent (ELISA) assay. In AAV overexpressing green fluorescent protein (AAV-GFP)-infected control cells, low-level NT-3 expression and baseline neurite outgrowth profiles (stained with βIII-tubulin) were observed (*Figure 1C*, arrows). In contrast, NT-3 expression and neurite outgrowth in AAV-NT-3-treated cultures were markedly increased (*Figure 1D*, arrows). AAV-NT-3 infection significantly enhanced NT-3 expression (*Figure 1E*, $t = 9.967$, df = 15, $p<0.0001$) and neurite length of spinal cord neurons (*Figure 1F*, $t = 4.273$, df = 73, $p<0.0001$), as compared to the AAV-GFP controls. At both 5 and 9 days *in vitro*, conditioned medium from the control spinal cord neurons treated with the phosphate buffered saline (PBS) or AAV-GFP had almost no detectable levels of NT-3, as assessed by ELISA (*Figure 1G*). However, cells receiving AAV-NT-3 demonstrated a significant increase in NT-3 protein expression as compared to the PBS and AAV-GFP groups at 5 days and 9 days in culture (Fig. 1G, $F_{2, 6} = 104.4$, $F_{2, 6} = 42.17$, $p<0.001$).

To test the effect of AAV-NT-3 on neurite outgrowth of spinal cord neurons in real time, we captured time-lapsed images of spinal cord neurons using an IncuCyte ZOOM Kinetic Imaging System. We measured the morphological parameters based on the contrast images of the spinal cord

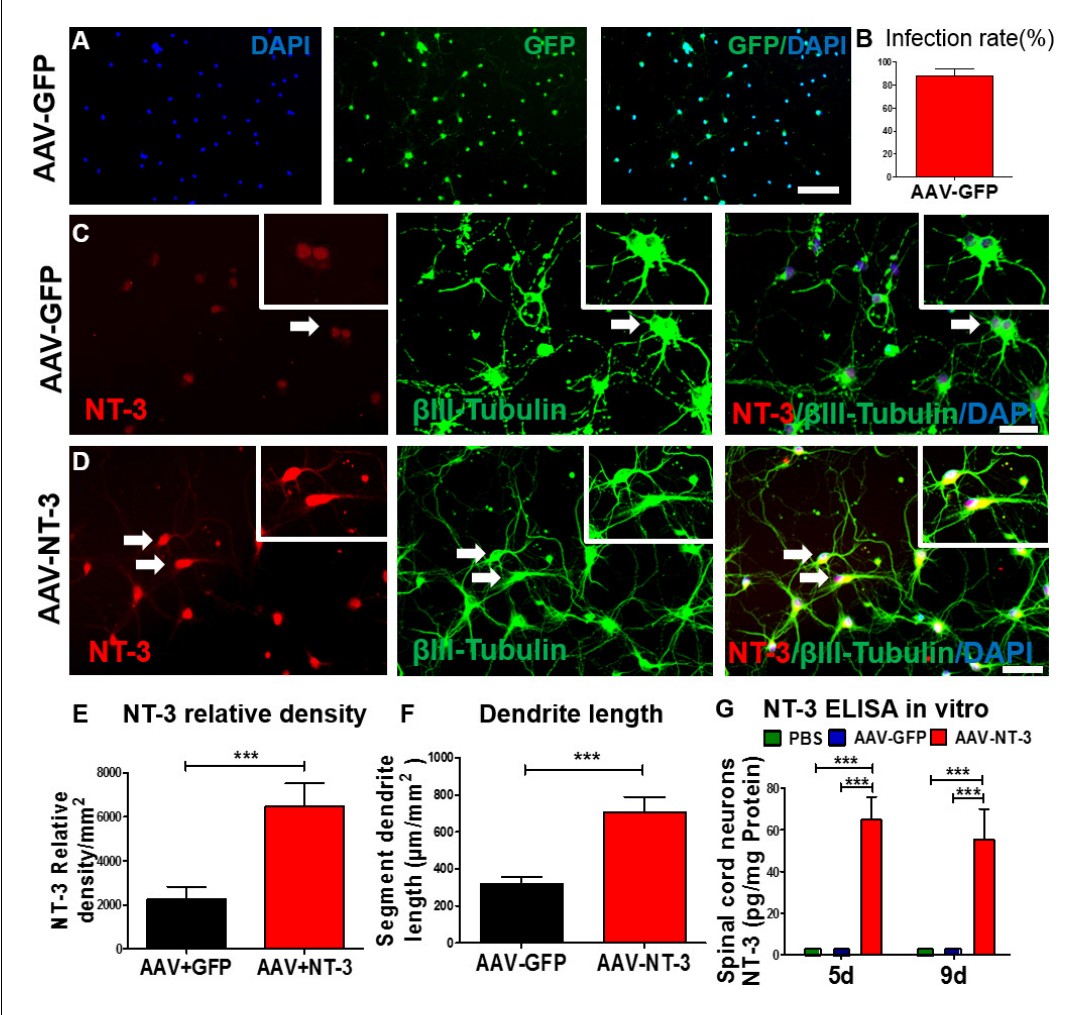

**Figure 1.** AAV-NT-3 enhanced NT-3 expression, secretion, and neurite extension in spinal cord neurons *in vitro*. (A) AAV-GFP infected spinal cord neurons (green) counterstained with DAPI (blue) *in vitro*. (B) AAV-GFP infection in spinal cord neurons was highly efficient (infection rate: 87.4 ± 6.34%). (C and D) Representative immunofluorescent staining shows triple labeling of NT-3 (red, arrows), βIII-Tubulin (green, for neuronal cell bodies and dendrites, arrows), and DAPI (blue, a nuclear dye) in spinal cord neurons in the AAV-GFP group (C) and AAV-NT-3 group (D) (n = 8 wells/group). (E and F) Quantitative analyses of NT-3 expression (E) and dendrite length (F) in spinal cord neurons based on immunofluorescent staining shown in C and D. (G) NT-3 concentration measured by ELISA in spinal cord neurons at 5 and 9 days in culture after addition of PBS, AAV-GFP and AAV-NT-3 (n = 3 mice/group). Error bars show mean ± SD. *** $p < 0.001$, *Student's t* tests, One-way ANOVA, Tukey's *post hoc* test. Scale bars: A, 100 μm; C and D, 50 μm. Abbreviations: AAV, adeno-associated virus (serotype 2); DAPI, 4',6-diamidino-2-phenylindole; ELISA, enzyme-linked immunosorbent assay; GFP, green fluorescent protein; NT-3, neurotrophin-3; PBS, phosphate buffered saline; SD, standard error of the mean.

DOI: https://doi.org/10.7554/eLife.39016.002

The following figure supplement is available for figure 1:

**Figure supplement 1.** AAV-NT-3 promoted neurite outgrowth *in vitro*.

DOI: https://doi.org/10.7554/eLife.39016.003

neurons treated with AAV-GFP (*Figure 1—figure supplement 1A*) and AAV-NT-3 (*Figure 1—figure supplement 1B*) acquired every 4 hr for 5 days. AAV-NT-3 dramatically promoted increases in neurite length (*Figure 1—figure supplement 1C*), neurite branch points (*Figure 1—figure supplement 1D*), neurite length/cell body (*Figure 1—figure supplement 1E*), and neurite branch/cell body of spinal cord neurons (*Figure 1—figure supplement 1F*). These results demonstrate that AAV-NT-3 transfection promoted NT-3 expression in spinal cord neurons resulting in robust neurite outgrowth of these neurons as compared to the controls transfected with AAV-GFP *in vitro*.

## Lysolecithin induced transient demyelination of the sciatic nerve facilitating AAV-NT-3 infection efficiency without causing prolonged neurological deficits

We previously demonstrated that lysolecithin-induced transient demyelination facilitated the infection of motoneurons and dorsal root ganglion (DRG) neurons following adenovirus injections into the sciatic nerve (*Zhang et al., 2010*). We tested whether the same strategy could enhance AAV-NT-3 infection in the sciatic nerve. We injected a mixture of AAV-GFP and AAV-NT-3 (ratio = 1:1) into the same site of sciatic nerves at 5 days after the lysolecithin injection (*Figure 2C*). Three days after the viral vector injection, mice were sacrificed and longitudinal-sections of the sciatic nerve were examined. P0 (a marker for peripheral myelin) staining showed that demyelination extended approximately 8.42 ± 2.22 mm proximodistally centered at the injection site (*Figure 2—figure supplement 1A*). Compared to the non-treated control group, the lysolecithin treatment significantly increased GFP expression (five-fold, $t = 5.826$, df = 8, $p<0.0001$; *Figure 2—figure supplement 1D*) and decreased P0 expression (6-fold, $t = 6.324$, df = 8, $p<0.0001$, *Figure 2—figure supplement 1E*). However, axons, identified by neurofilament 200 (NF200) immunoreactivity (IR), remained intact and were not affected by lysolecithin-induced demyelination ($t = 1.065$, df = 8, ns, compared to the PBS-injected control, *Figure 2—figure supplement 1A,I*). Five weeks after viral vector injection, no significant differences in P0 expression were found between lysolecithin injected and non-injected groups ($t = 0.711$, df = 8, ns, *Figure 2—figure supplement 1F,J*). However, in the lysolecithin-injected group that received an AAV-NT-3 infection, a significant increase in NT-3 expression in the sciatic nerve was found as compared to the PBS injected group ($t = 5.282$, df = 8, $p<0.001$, *Figure 2—figure supplement 1K*). Notably, Basso Mouse Scale (BMS) locomotor test showed transient locomotor deficit at 1 day post lysolecithin injection ($t = 3.841$, df = 8, $p<0.01$), but was quickly returned to control levels at 7 and 14 days post-injection (7d: $t = 2.86$, df = 8, ns; 14d: $t = 1.342$, df = 8, ns; *Figure 2—figure supplement 1L*). These results collectively indicate that lysolecithin injection induced transient demyelination which effectively eliminated the infection barrier formed by the myelin sheath and facilitated AAV-NT-3 entry into the sciatic nerve axons. It is also important to note that the lysolecithin injection did not damage passing axons and no animals sustained any prolonged neurological deficits.

## AAV-NT-3 was retrogradely transported to lumbar motoneurons, expressed, and released by transduced motoneurons after the sciatic nerve transfection

To verify the retrograde transport of AAV from the site of sciatic nerve injection to the lumbar spinal cord MNs and their NT-3 expression, AAV-GFP and AAV-NT-3 were mixed (1:1 ratio) and injected into the transiently demyelinated sciatic nerves as described above. Five weeks later, immunostaining of transverse sections of the lumbar spinal cord ventral horn revealed AAV-GFP-labeled MNs (*Figure 3A*) as well as DRG axons innervating the dorsal gray matter (*Figure 3A* and *Figure 3—figure supplement 1A*). Within the L2-L5 DRGs, GFP expression was detected in the neuronal somata (*Figure 3—figure supplement 1B*), confirming the specificity of AAV-GFP expression in MNs, DRG neurons, and sensory axons (*Zhang et al., 2010*).

As anticipated, GFP expression was present in motoneurons in both the AAV-GFP and AAV-GFP/AAV-NT-3 mixed injection groups between the L2-L5 spinal cord segments (*Figure 3B and C*), indicating that the GFP and NT-3 molecules were retrogradely transported from the site of sciatic nerve injection to these lumbar motoneurons. NT-3 immunofluorescent density further revealed that the expression of NT-3 in the AAV-NT-3 injection group was significantly higher than that the AAV-GFP injection group ($t = 6.695$, df = 10, $p<0.001$, *Figure 3D*). Increased NT-3 in the L2-L5 lumbar spinal cord after the AAV-NT-3 transfection was also confirmed by the ELISA assay. The concentration of NT-3 in the lumbar spinal cord after sciatic nerve infection of AAV-NT-3 showed an 8-fold increase in the amount of NT-3 as compared to the AAV-GFP infection ($t = 27.79$, df = 4, $p<0.001$, *Figure 3E*). Thus, our data indicate that biologically active NT-3 can be retrogradely transported to lumbar motoneurons, and can be expressed and released by transduced motoneurons in the lumbar spinal cord after their retrograde transport. Our results were in agreement with a previous report using a similar approach (*Zhang et al., 2010*).

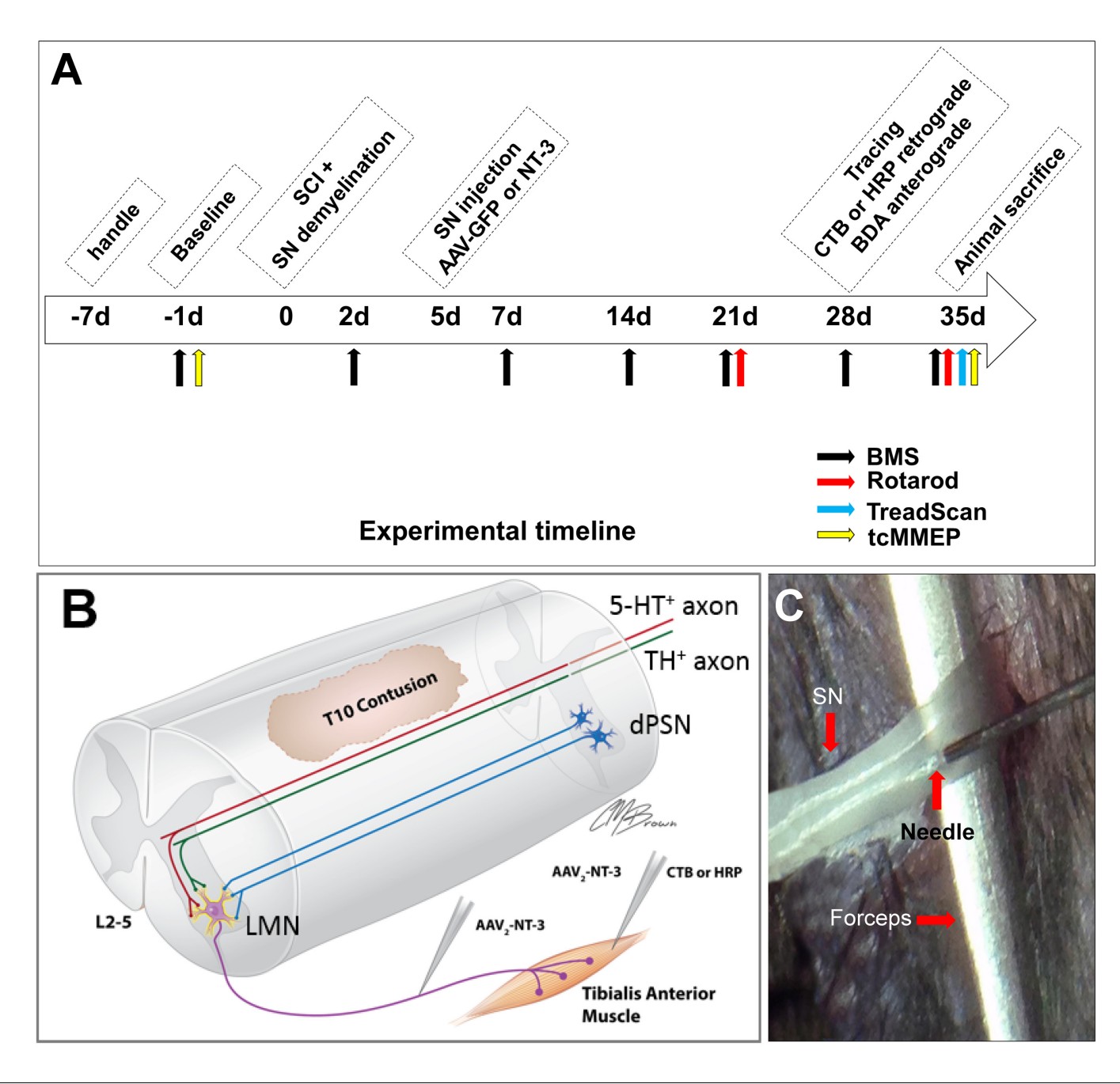

**Figure 2.** *In vivo* Experimental Design. (**A**) Experimental timeline. (**B**) Schematic drawing shows the experimental design. A spinal cord contusion injury was made at T10. AAV-NT-3 was injected into the sciatic nerve to allow retrograde transport of NT-3 to lumbar MNs. Retrograde tracers CTB or HRP were injected into the Tibialis Anterior muscle. Descending serotonergic (5-HT⁺), dopaminergic (TH⁺), and propriospinal (dPST) axons were examined in their sprouting and synaptogenesis on lumbar MNs. (**C**) Photograph of the sciatic nerve injection. The exposed nerve was loosely held with arch-tipped forceps to prevent lateral nerve movement. Intraneural injection was performed using a 32-gauge needle attached to a Hamilton syringe.
Abbreviations: 5-HT, 5-hydroxytryptamine; AAV, adeno-associated virus (serotype 2); BDA, biotinylated dextran amine; BMS, Basso Mouse Scale; CTB, cholera toxin B; d, day; HRP, horseradish peroxidase; L, lumbar; LMN, lumbar motoneurons; NT-3, neurotrophin-3; SCI, spinal cord injury; SN, Sciatic nerve; tcMMEP, transcranial magnetic motor-evoked potentials; TH, tyrosine hydroxylase.
DOI: https://doi.org/10.7554/eLife.39016.004
The following figure supplement is available for figure 2:

**Figure supplement 1.** Transient demyelination of sciatic nerves induced by lysolecithin injection.
*Figure 2 continued on next page*

*Figure 2 continued*

DOI: https://doi.org/10.7554/eLife.39016.005

## AAV-NT-3 infection facilitated behavioral and electrophysiological recoveries after a moderate contusive SCI in adult mice

A standardized 0.5 mm displacement contusion at 10th thoracic vertebral level (T10) induced a consistent moderate SCI in mice. No significant differences in the force, velocity, or injury time were observed among the control and treatment groups (*Figure 4—figure supplement 1*). Using this model and combined behavioral (BMS, rotarod, TreadScan) and electrophysiological (tcMMEP) assessments, we determined whether AAV-NT-3 treatment resulted in improved functional

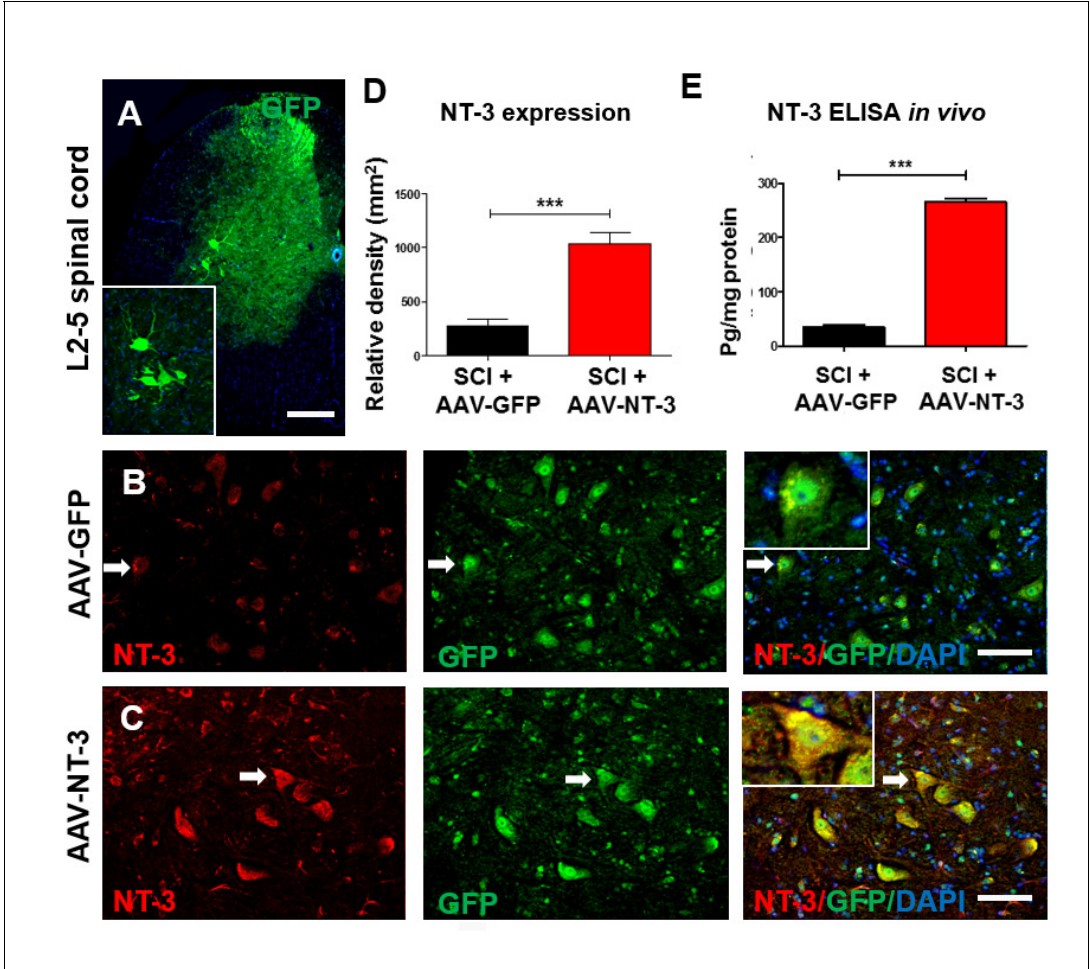

**Figure 3.** Retrogradely transported AAV-NT-3 enhanced NT-3 expression in lumbar motoneurons. (A) A photomicrographic image of a lumbar spinal cord (L2-5) cross section shows AAV-GFP labeled ventral horn motoneurons (also seen in the insert) and dorsal horn afferent sensory axons. (B and C) Immunofluorescent triple staining of NT-3 (red, arrows), GFP (green, arrows) and DAPI (blue) in the AAV-GFP (B) and AAV-NT-3 (C) groups. (D and E) Comparison in NT-3 relative density (D) and expression (E) between the AAV-NT-3 and AAV-GFP groups (n = 3 mice/group). Error bars show mean ± SD. ***$p < 0.001$, *Student's t* tests. Scale bars: A, 200 μm; B, C, 50 μm. Abbreviations: AAV, adeno-associated virus (serotype 2); DAPI, 4',6-diamidino-2-phenylindole; ELISA, enzyme-linked immunosorbent assay; GFP, green fluorescent protein; NT-3, neurotrophin-3; SD, standard error of the mean.

DOI: https://doi.org/10.7554/eLife.39016.006

The following figure supplement is available for figure 3:

**Figure supplement 1.** Expression of AAV-GFP in dorsal root axons and DRG neurons, and co-localization of ChAT and CTB in lumbar MNs.

DOI: https://doi.org/10.7554/eLife.39016.007

recoveries as compared to the AAV-GFP control (timeline, *Figure 2*). We found that AAV-NT-3 treatment resulted in a significant and progressive recovery of locomotion as compared to the AAV-GFP control group (*Figure 4A*). Repeated measures analysis of variance (ANOVA) revealed that the BMS scores at 4 and 5 weeks post-SCI in the AAV-NT-3 treated group were significantly higher than the AAV-GFP group (two-way ANOVA, time: $F_{6, 162}$ = 245.2, p<0.001; column factor: $F_{2, 27}$ = 298.3, p<0.001; *Figure 4A and B*). Significant effect of AAV-NT-3 treatment was found on the rotarod run time at 5 weeks post-SCI as compared to the control group receiving AAV-GFP ($F_{2, 27}$ = 28.16, p<0.001; *Figure 4C and D*). TreadScan analysis (*Figure 4E–H*) demonstrated that administration of AAV-NT-3 significantly decreased footprint angle ($F_{2, 27}$ = 9.19, p<0.001, *Figure 4F*) and improved footprint length ($F_{2, 27}$ = 15.67, p<0.001, *Figure 4G*) and toe spread ($F_{2, 27}$ = 9.45, p<0.001, *Figure 4H*) at 5 weeks post-SCI.

Electrophysiologically, transcranial magnetic motor evoked potentials (tcMMEP) recordings revealed three responses: (1) no response (NR, *Figure 5D*), early response (ER, *Figure 5E*), and late response (LR, *Figure 5F*); AAV-NT-3 treatment increased the number of ER as compared to the AAV-GFP control ($\chi^2$=0.833, p<0.001, *Figure 5B*). In both groups, the amplitudes of tcMMEP were significantly reduced, as compared to the sham group (*Figure 5C*). However, compared to the AAV-GFP group, the AAV-NT-3 group demonstrated significantly increased amplitude (ER: $F_{2, 15}$ = 15.67, p<0.001; LR: $t$ = 7.630, df = 10, p<0.001, *Figure 5G*) and decreased latency (ER: $F_{2, 15}$ = 16.55, p<0.001; LR: $t$ = 3.993, df = 10, p<0.01, *Figure 5H*) in early and late responses as compared to the AAV-GFP group after SCI. Thus, both behavioral and electrophysiological results indicate the positive influence of NT-3 on recovery of motor function after SCI.

## AAV-NT-3 infection had no effect on lumbar motoneuron number and somal size

To determine whether AAV-NT-3 infection affected MN number and somal size, we injected horseradish peroxidase conjugated to the cholera toxin B subunit (BHRP) into the anterior tibialis (TA) muscle and showed successful BHRP labeling in motoneurons in all groups. BHRP-labeled MNs were located in the lateral motor column of the L2-L5 spinal segments and their dendritic arbors were strictly unilateral with extensive ramification along the ventrolateral edges of the gray matter, the lateral funiculus, and throughout the ventral horn (*Figure 6A*). The number of MNs per animal labeled with BHRP ($F_{2, 21}$ = 0.517, ns, *Figure 6—figure supplement 1C*) or stained with Thionin ($F_{2, 15}$ = 0.541, ns, *Figure 6—figure supplement 1B*) did not differ across groups (*Figure 6A*, *Figure 6—figure supplement 1A*). The area of MN somata also did not differ across treatment groups ($F_{2, 21}$ = 1.647, ns, *Figure 6—figure supplement 1D*). These results indicate that a thoracic contusive SCI did not cause lumbar MN loss and that AAV-NT-3 infection did no affect MN somal size.

## AAV-NT-3 infection attenuated motoneuron dendritic atrophy

To determine whether AAV-NT-3 had any effect on dendritic morphology, the BHRP-labeled dendritic arbors of TA MNs were measured by computer-generated composites and polar histograms (*Figure 6A*). Following the T10 contusive SCI which did not affect lumbar MNs directly, the dendritic length per lumbar TA MNs was significantly decreased. However, treatment with AAV-NT-3 significantly attenuated SCI-induced MN dendritic atrophy ($F_{2, 15}$ = 47.96, p<0.001, *Figure 6C*).

In the sham control, length per radial bin of TA MN dendrites displayed a non-uniform distribution, between 300 ° and 120 °. SCI induced a decrease in MN dendritic length in specific radial bins (*Figure 6A and B*). Treatment with AAV-NT-3 significantly attenuated SCI-induced reduction in MN dendritic length in specific radial bins (0 – 60 °, $F_{2, 15}$ = 61.07, p<0.001; 60 – 120 °, $F_{2, 15}$ = 11.59, p<0.001; 300 – 360 °, $F_{2, 15}$ = 13.48, p<0.001, *Figure 6B*). Notably, there were strong correlations between the MN dendrite length per cell and both the BMS ($r$ = 0.87, p<0.001, *Figure 6D*) and rotarod ($r$ = 0.71, p<0.001, *Figure 6E*) of behavioral assessments, indicating that motor functional improvements were likely influenced by increased MN dendritic arbors.

## AAV-NT-3 infection enhanced descending serotonergic, dopaminergic and propriospinal innervations of the lumbar MN pool

The mesencephalic dopaminergic neurons represent a vital neuromodulatory component essential for vertebrate motor control (*Ryczko et al., 2016*). Descending dopaminergic neurons are located in

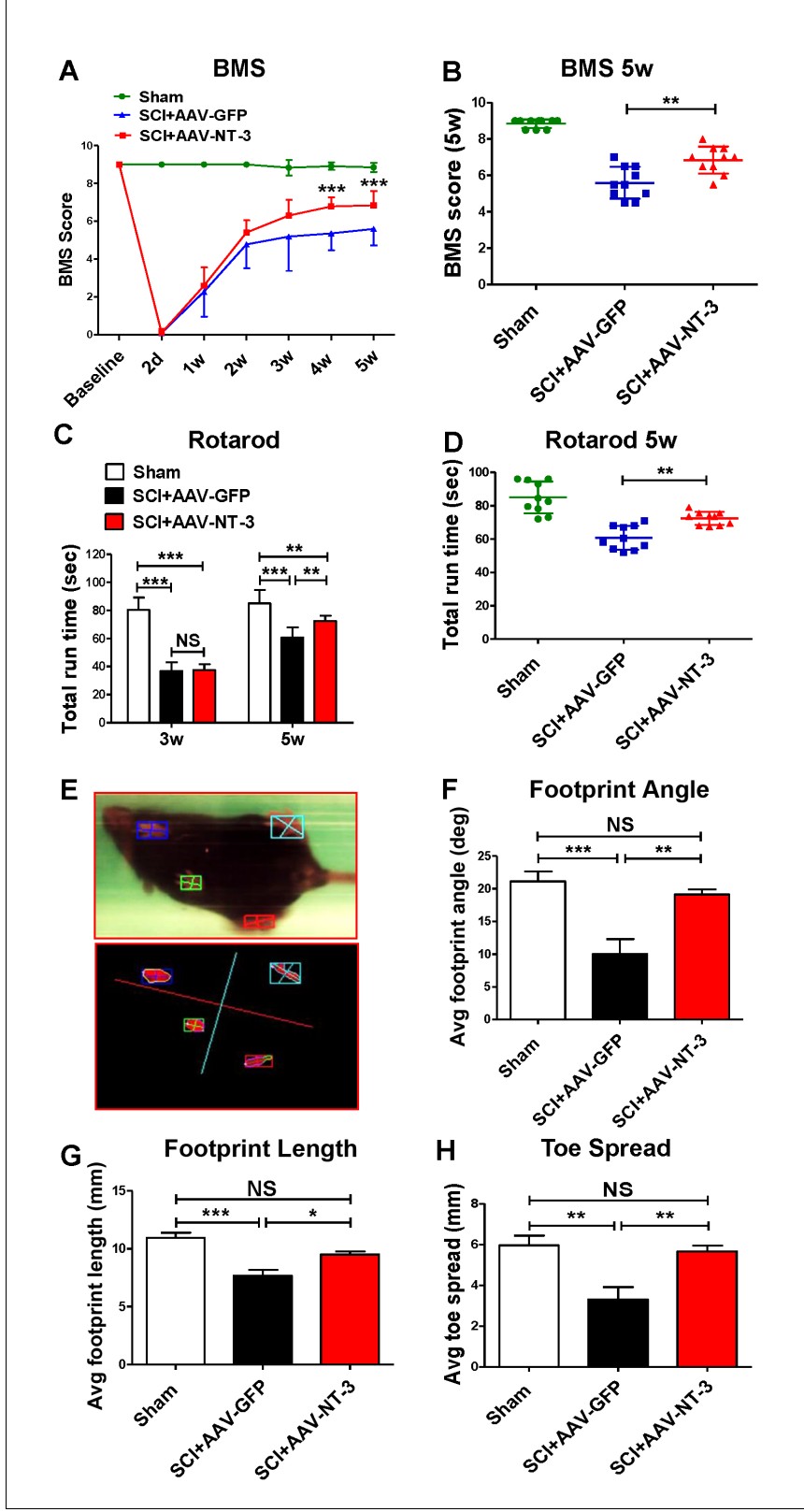

**Figure 4.** Retrogradely transported AAV-NT-3 enhanced hindlimb locomotor recovery. (**A and B**) AAV-NT-3 significantly improved Basso Mouse Scale (BMS) for locomotion at 4 and 5 weeks post-SCI. (**C and D**) AAV-NT-3 significantly increased rotarod run times at 5 weeks post-injury as compared to AAV-GFP. (**E – H**) TreadScan analyses show captured footprint images (**E**), and quantifications of footprint angle (**F**), footprint length (**G**), and

*Figure 4 continued on next page*

*Figure 4 continued*

toe spread (**H**). n = 10 mice/group, error bar = mean ± SD. **$p < 0.01$, ***$p < 0.001$, repeated measures two-way ANOVA, Tukey's post-hoc test (BMS), One-way ANOVA, Kruskal–Wallis post-hoc test (rotarod and TreadScan). Abbreviations: AAV, adeno-associated virus (serotype 20; GFP, green fluorescent protein; NS, no significance; NT-3, neurotrophin-3; SCI, spinal cord injury; SD, standard error of the mean.
DOI: https://doi.org/10.7554/eLife.39016.008
The following figure supplement is available for figure 4:

**Figure supplement 1.** Spinal cord injury parameters produced by an Infinite Horizon injury device.
DOI: https://doi.org/10.7554/eLife.39016.009

the A8, A9 and A10 groups of the mesencephalon and express the transmitter synthesizing enzyme tyrosine hydroxylase (TH) (*Zaborszky and Vadasz, 2001*). NT-3 has a wide range of effects on survival, biosynthetic activities and neuroprotective roles of dopaminergic neurons, likely via autocrine or paracrine mechanisms (*Hyman et al., 1994*; *Seroogy et al., 1994*). In the spinal cord neuropil, TH-IR axons are of supraspinal origin and project to and beyond the lumbar spinal cord. We found a significant reduction of TH-IR dopaminergic axons innervating the lumbar ventral horn neuropil after a contusive SCI at T10 (*Figure 7A*). Importantly, AAV-NT-3 infection resulted in a significant increase in the TH-IR axon density in the lumbar neuropil as compared to the AAV-GFP ($F_{2, 15} = 7.59$, p<0.001, *Figure 7A and D*).

The descending serotonergic pathway plays an important role in mediating descending influences on locomotion (*Bowker et al., 1983*). Axons of this pathway originate mainly from the nucleus raphe obscurus, the nucleus raphe pallidus, and the nucleus raphe magnus of the caudal brainstem, express the transmitter serotonin (5-HT), and project to and beyond the lumbar spinal cord. After a contusive SCI at T10, descending serotonergic projections were partially damaged at the lesion site, but spared axons projected to the lumbar spinal cord and beyond. Indeed, we found a significant reduction of 5-HT-IR axon terminals in the lumbar ventral horn MN area (*Figure 7B*). Notably, AAV-NT-3 infection resulted in an about 1.5-fold increase in the 5-HT-IR axon density in the lumbar neuropil as compared to the AAV-GFP control group ($F_{2, 15} = 8.50$, p<0.01, *Figure 7B and E*).

The descending propriospinal tract (dPST) axons also play an essential role in mediating motor function. Following SCI, the dPST may serve as a functional relay conveying supraspinal motor command down to the distal spinal cord (*Courtine et al., 2008*; *Deng et al., 2013*). To trace the course of dPST axons around the lesion and down to the lumbar motoneuron pool, we injected an anterograde tracer biotinylated dextran amine (BDA) into the intermediate gray matter of the T7-T8 cord segment rostral to a T10 contusion. We found that spared BDA-labeled dPST axons extended around the lesion through spared white matter (*Figure 7—figure supplement 1C*). At the lumbar spinal cord level, many BDA-labeled dPST axons were found in the ventral horn MN pool. Numerous dPST axon terminals formed contacts (*Figure 7C*) or synapse-like contacts (*Figure 7—figure supplement 1A,B*) on MNs at L2-L5 spinal cord levels. The number of dPST axons co-labeled with MNs/dendrites (*Figure 7—figure supplement 1A*) and synapse-like contacts (*Figure 7—figure supplement 1B*) were increased in the AAV-NT-3 treated animals relative to the non-treated controls (statistical data not shown). Consistent with 5-HT-IR and TH-IR axons, AAV-NT-3 treatment significantly increased the density of dPST axons in the lumbar MN pool as compared to the non-treated control group ($F_{2, 15} = 8.89$, p<0.01, *Figure 7C and F*). These results collectively indicate that AAV-NT-3 treatment enhanced synaptic inputs of descending supraspinal (serotonergic, dopaminergic) and propriospinal (dPST) axons to lumbar MNs that remained intact but underwent atrophy and synaptic stripping after a thoracic contusive SCI.

## AAV-NT-3 infection enhanced synaptic contacts on lumbar MNs and their dendrites

To quantify the degree of synaptic stripping and reconnection, cholera toxin B subunit (CTB) was injected into the sciatic nerves to retrogradely label lumbar MNs at 28 days post AAV-NT-3 or AAV-GFP gene transfer. As anticipated, CTB was co-labeled with choline acetyltransferase (ChAT) in the lumbar ventral horn MNs and the CTB-labeled MN dendrites was more robust than the ChAT labeling (*Figure 3—figure supplement 1C–G*).

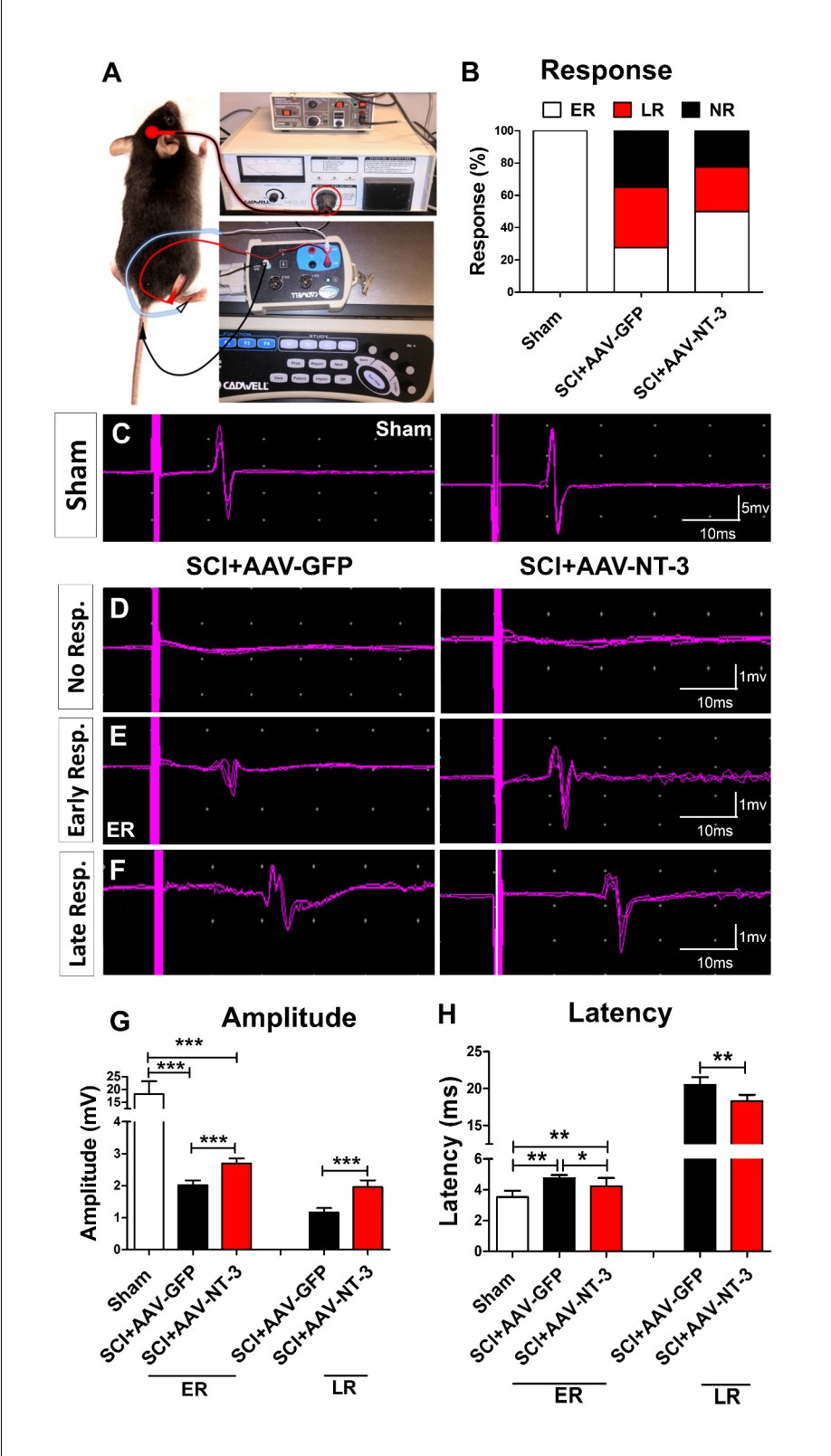

**Figure 5.** Electrophysiological measure with transcranial magnetic motor-evoked potentials (tcMMEP). (**A**) tcMMEP set up. (**B**) Compared to the sham group, three types of tcMMEP responses, i.e. early response (ER), late response (LR), and no response (NR) were observed. AAV-NT-3 markedly increased the percent of early response. (**C**) Examples of normal response waveforms in the sham group. (**D–F**) Examples of no response (**D**), early response (**E**), and late response (**F**) waveforms in the AAV-GFP and AAV-NT-3 treatment groups at 5 weeks after SCI. (**G and H**) Compared to the

*Figure 5 continued on next page*

*Figure 5 continued*

control groups, AAV-NT-3 significantly increased the amplitude (**G**) and decreased the latency (**H**). n = 10 mice/group. $^*p < 0.05$, $^{**}p < 0.01$, $^{***}p < 0.001$, $\chi^2$ test (early response), One-way ANOVA, Kruskal–Wallis post-hoc; *Student's t* tests (late response).

DOI: https://doi.org/10.7554/eLife.39016.010

Next, we used an automatic detection method to determine the density of synaptophysin (SYP, a major synaptic marker) in a perimeter of 5 µm-wide area surrounding the MN soma (*Figure 8—figure supplement 1A*, *Figure 8A*). We found that the SYP density around ChAT-labeled MNs in the SCI + AAV-NT-3 group was significantly higher than that of the SCI + AAV GFP group ($t = 2.63$, p<0.05, *Figure 8—figure supplement 1B–C*). In cross sections of the lumbar spinal cord, SYPs surrounding the MNs were decreased in the SCI + AAV GFP group as compared to the sham control group. AAV-NT-3 treatment prevented this decrease in SYP relative density ($F_{2, 21} = 12.46$, p<0.001, *Figure 8A–B*), increased the SYP density per MN soma per section ($F_{2, 33} = 20.83$, p<0.001, *Figure 8C*) as well as SYP density per dendritic segment per section ($F_{2, 21} = 7.65$, p<0.01, *Figure 8D*). Taken together, these results suggest that SCI induced reduction of synaptic contacts on motoneurons and their dendrites, and that NT-3 overexpression by MNs and their dendrites may enhance the synaptic formation or attenuated the loss of synapses on MNs and their dendrites.

## AAV-NT-3 infection significantly reduced tibialis anterior muscle atrophy but had no effect on motor endplate density

To assess whether AAV-NT-3 infection affect TA muscle atrophy, we scaled the TA muscle weight. Following SCI in the AAV-GFP group, TA muscle weight decreased by 33.4% as compared to the sham group, indicating that SCI induced TA muscle atrophy. However, the TA muscle weight in the AAV-NT-3 infected group increased 22.5% as compared to the AAV-GFP treated group after SCI ($F_{2, 15} = 29.96$, p<0.001, *Figure 9C*). TA muscle fiber diameter in the SCI + AAV GFP group was also significantly reduced as compared to the sham group (*Figure 9A and D*). Notably, AAV-NT-3 treatment significantly increased fiber diameter as compared to the AAV-GFP group following SCI ($F_{2, 15} = 35.15$, p<0.001, *Figure 9D*).

We also examined the density of motor endplates and found that they were unaffected by the contusive SCI in different treatment groups ($F_{2, 21} = 0.465$, ns, *Figure 9B and E*). Collectively, these results indicate that AAV-NT-3 infection significantly reduced TA muscle atrophy as compared to the AAV-GFP following SCI, with increased TA muscle weights and fiber diameters compared to the untreated controls.

## AAV-NT-3 infection had no effect on the lesion size and myelin sparing at the site of a T10 contusive SCI

To assess whether AAV-NT-3 infection of lumbar motoneurons had any effect on lesion volume and tissue sparing at the site of contusion injury, we measured the percentage of lesion area in sections stained with Cresyl Violet and Eosin. Results showed that AAV-NT-3 treatment had no effect on the percentages of lesion area ($t = 0.689$, df = 10, ns, *Figure 9—figure supplement 1A,C*) and lesion volume ($t = 0.073$, df = 10, ns, *Figure 9—figure supplement 1D*) at the lesion site as compared to the AAV-GFP group. Similarly, Luxol Fast Blue staining showed no difference in the percent of myelin sparing area between the two groups ($t = 0.255$, df = 10, ns, *Figure 9—figure supplement 1B,E*) indicating that local infection of lumbar MNs had no effect on the lesion size and myelin sparing rostral to the lumbar neuropil.

## Discussion

The goal of this study was to determine whether targeted delivery of NT-3 to lumbar MNs caudal to a T10 contusive SCI could modulate lumbar neural circuitry, leading to functional recovery. *In vitro* studies demonstrated that AAV-NT-3 infection enhanced NT-3 expression on cultured spinal cord neurons and promoted dendrite outgrowth. *In vivo*, we showed that AAV-NT-3, after targeted delivery into transiently demyelinated sciatic nerves, was retrogradely transported to and released from lumbar MNs, contributing to the remodeling of lumbar motoneuron circuits and attenuation of SCI-

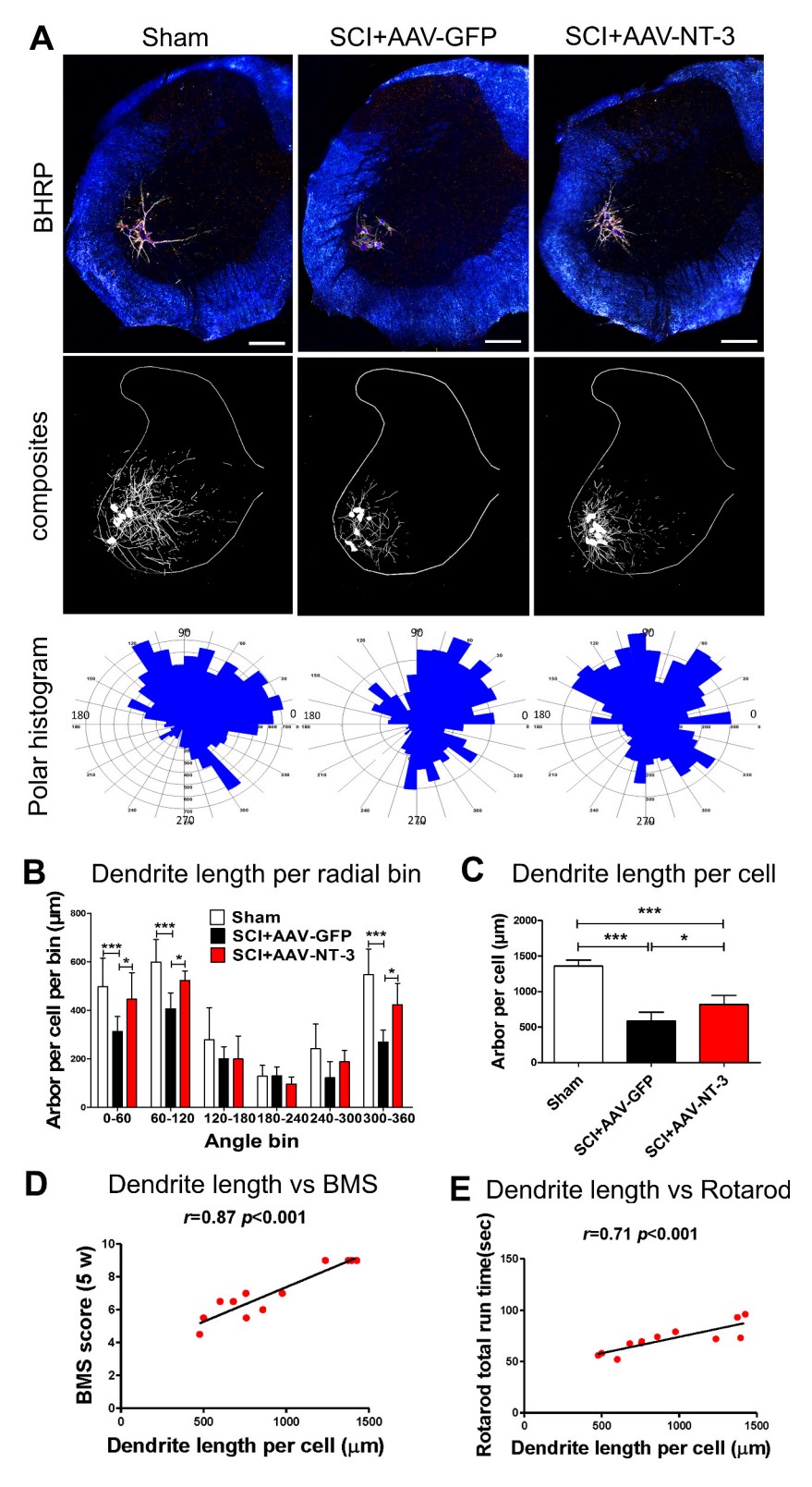

**Figure 6.** Retrogradely transported AAV-NT-3 attenuated dendritic atrophy. (**A**) Dark-field micrographs and matching computer-generated composites of transverse hemisections through the lumbar spinal cords of sham, SCI + AAV GFP, and SCI + AAV-NT-3 groups after BHRP injection into the tibialis anterior (TA) muscle. Computer-generated composites of BHRP-labeled somata and processes were drawn at 480 μm intervals through the entire rostrocaudal extent of the TA motor pool. Polar histogram of BHRP-labeled TA motoneuron dendrites was measured. (**B**) Length per radial bin of TA

*Figure 6 continued on next page*

*Figure 6 continued*
dendrites displayed a non-uniform distribution, with the majority of the arbor located between 300 ∘ and 120 ∘. Following SCI, dendritic lengths of these radial bins in the SCI + AAV GFP group reduced significantly. Treatment with $AAV_2$-NT-3 attenuated these reductions. (C) The dendritic length per TA motoneuron was significantly reduced in the AAV-GFP group and treatment with AAV-NT-3 significantly attenuated this dendritic atrophy. (D and E) Strong correlation between dendrite length per cell and BMS (D) and rotarod (E) scores for sham, SCI + AAV GFP and SCI + AAV-NT-3 groups (n = 4/group). Error bars show mean ±SD. $^*p < 0.05$, $^{***}p < 0.001$, One-way ANOVA, Kruskal–Wallis post-hoc (B, C); Pearson's correlation coefficient (*r* value, (D, E). n = 4 mice/group. Scale bar in A, 200 µm.
DOI: https://doi.org/10.7554/eLife.39016.011
The following figure supplement is available for figure 6:

**Figure supplement 1.** Motoneuron counts.
DOI: https://doi.org/10.7554/eLife.39016.012

induced lumbar MN dendritic atrophy and muscle atrophy. Moreover, NT-3 enhanced innervation and synaptogenesis of several descending axonal pathways on lumbar MNs. These morphological changes correlate well with behavioral and electrophysiological improvements after targeted delivery of NT-3. Thus, the work described herein is, to the best of our knowledge, the first demonstration of NT-3 as an effective regulator of lumbar spinal circuit caudal to an SCI, supporting a role for retrograde transport of NT-3 as a potential therapeutic strategy for SCI.

## AAV-NT-3 effectively infected spinal neurons and promoted NT-3 expression *in vitro*

For clinical relevance, we used an AAV-mediated gene transfer technique to deliver NT-3. The AAV-mediated gene delivery has been reported as the most promising therapeutic transgene delivery system for prolonged, safe, and effective neurotrophin delivery (*Fortun et al., 2009*; *Petruska et al., 2010*; *Peng et al., 2011*). Earlier investigators showed an effect of NT-3 on dendrite morphogenesis in brain slice cultures and in proprioceptive axon patterning (*Ernfors et al., 1994*; *McAllister et al., 1995*; *Patel et al., 2003*). Here we confirmed that the AAV serotype two virus could infect cultured spinal cord neurons effectively and promoted NT-3 expression in these neurons. The time-lapse study revealed that AAV-NT-3 promoted dendrite outgrowth and branching in these neurons, which is in accord with a previous report (*Joo et al., 2014*).

## Retrogradely transported AAV-NT-3 enhanced NT-3 expression in transduced lumbar MNs

We and others have reported that myelin provides a protective barrier to the nervous system that limits AAV viral infection (*Zhou et al., 2003*; *Zhang et al., 2010*; *Hollis and Tuszynski, 2011*). Notably, lysolecithin-induced transient demyelination significantly increases the transduction efficiency of AAV. Compared with direct injection of AAV-NT-3 into the lumbar MN region, which may cause direct damage and inflammation at the injection site, the sciatic nerve injection eliminates such damage and allows transducing a gene to specific populations of neurons. In the present study, AAV-NT-3 was injected into demyelinated sciatic nerves 5 days after SCI, a clinically-feasible timeframe. Our results clearly demonstrated that lysolecithin-induced transient demyelination could effectively reduce the myelin sheath barrier and facilitate AAV-NT-3 entry, simultaneously avoiding excessive inflammation, secondary cell death, vascular disruption, infection of other cell types (e.g., glial cells), and the release of inhibitory molecules in the lumbar spinal cord (*Hollis and Tuszynski, 2011*). After the transient demyelination, axons at the injection site remained intact. BMS results showed minor impairment immediately after lysolecithin injection, which was rapidly normalized by 7 days post-injection. Immunofluorescence staining confirmed the specificity of GFP expression in lumbar MNs as well as in DRG neurons and sensory axons after transient demyelination and AAV-GFP infection, agreeing with a previous report (*Zhang et al., 2010*). ELISA analysis showed increases in NT-3 in the lumbar cord in the group receiving AAV-NT-3 compared to the AAV-GFP at 5 weeks after SCI. These results support the notion that biologically active NT-3 can be expressed and released by transduced MNs in the lumbar spinal cord, indicating an effective transfection and retrograde transport of AAV-NT-3 after transient demyelination of the sciatic nerves.

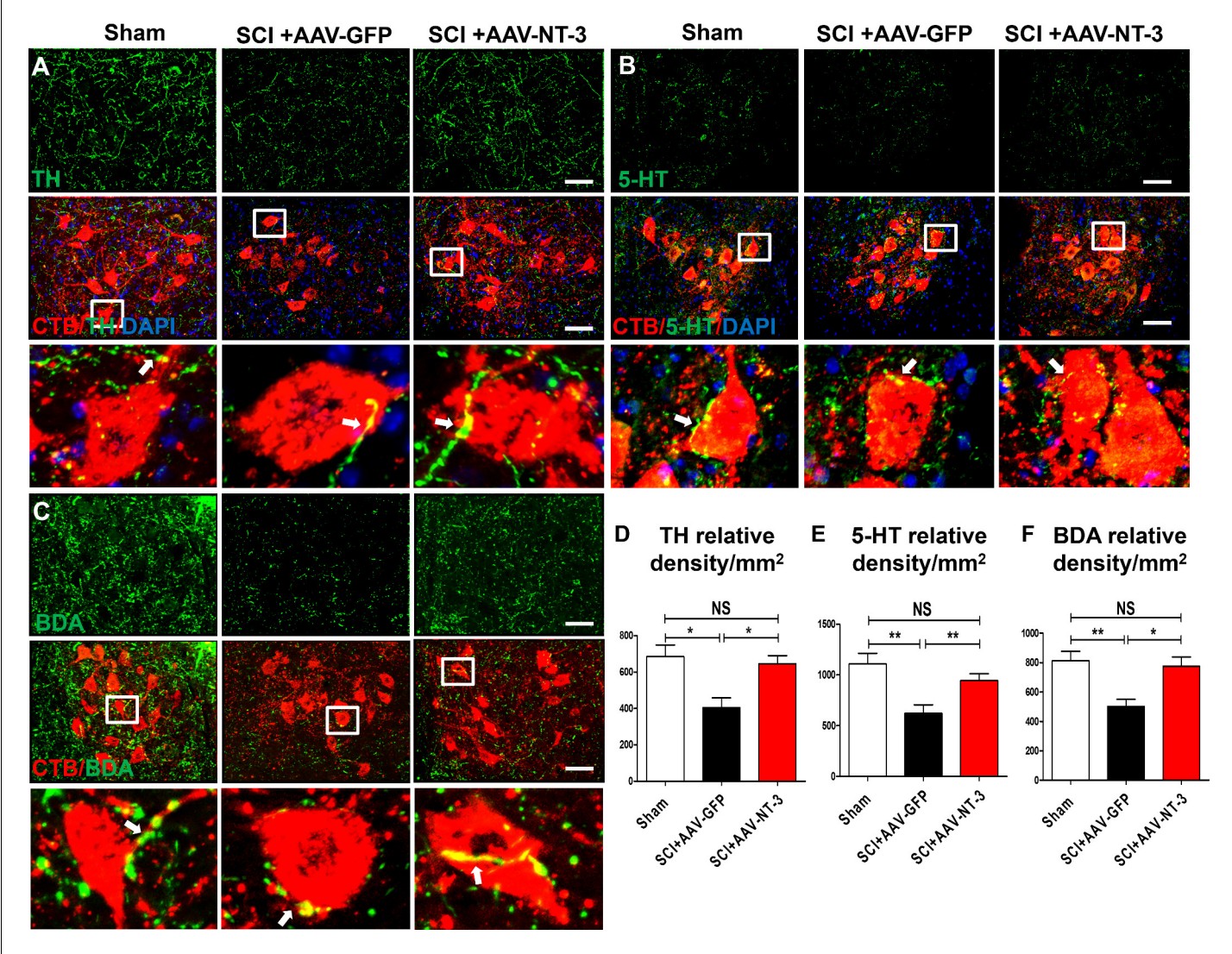

**Figure 7.** Retrogradely transported AAV-NT-3 enhanced terminal innervation by descending dopaminergic, serotonergic, and propriospinal axons in the lumbar motoneuron pool. (A–C) Representative images for TH+ dopaminergic axons (A, green), 5-HT+ serotonergic axons (B, green) and BDA+ dPST axons (C, green) in the lumbar motoneuron (MN) pool of different treatment groups at 5 weeks post-SCI. The lumbar MNs were retrogradely labeled with CTB (red) and counterstained with DAPI (a nuclear dye, blue). Inserts are high magnifications of boxed areas show synaptic-like structures established between TH-, 5-HT-, and dPST-labeled axons and CTB-labeled MNs (A-C, arrows). (D–F) Quantitative analyses of the TH+ relative integrated optical density (IOD)/mm$^2$ (D), 5-HT+ IOD (E), and BDA+ IOD (F). n = 6 mice/group. Error bars, mean ± SD. *p<0.05, **p<0.01, One-way ANOVA, Kruskal–Wallis post-hoc. Scale bars, 50 μm; Box scale bars, 20 μm. Abbreviations: 5-HT, 5-hydroxytryptamine or serotonin; AAV, adeno-associated virus (serotype 2); BDA, biotinylated dextran amine; CTB, cholera toxin B subunit; DAPI, DAPI dihydrochloride; GFP, green fluorescent protein; NS, no significance; NT-3, neurotrophin-3; SCI, spinal cord injury; SD, standard error of the mean.

DOI: https://doi.org/10.7554/eLife.39016.013

The following figure supplement is available for figure 7:

**Figure supplement 1.** Retrogradely transported AAV-NT-3 enhanced sprouting of dPST axons on lumbar MNs and formation of synaptic-like contacts on lumbar MNs.
DOI: https://doi.org/10.7554/eLife.39016.014

## Retrogradely transported AAV-NT-3 promoted behavioral and electrophysiological recoveries

Our results showed a beneficial effect of retrogradely transported AAV-NT-3 on functional recovery after SCI. AAV-NT-3 treatment significantly promoted BMS locomotor recovery at 4 and 5 weeks

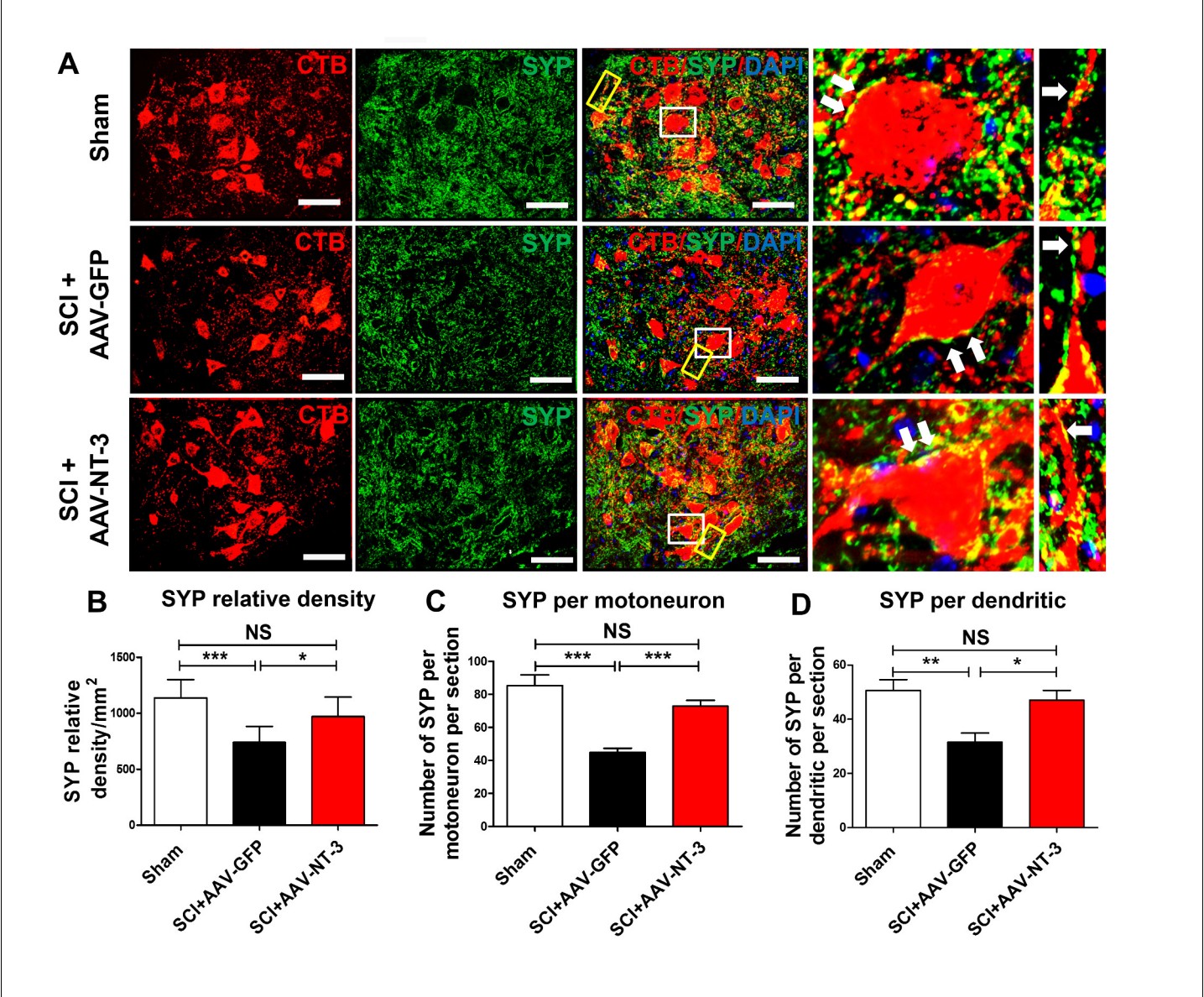

**Figure 8.** Retrogradely transported AAV-NT-3 enhanced synaptic density in the lumbar motoneuron pool. (A) Confocal images show triple staining of CTB-labeled MNs (red), synaptophysin (SYP, a presynaptic marker, green), and DAPI (a nuclear dye, blue) in the lumbar MN pool of 3 treatment groups at 5 weeks post-SCI. High magnifications of boxed areas depict co-localization between SYP+ presynaptic terminals and lumbar MNs (white box, arrows) and their dendrites (yellow box, arrows), indicating of synaptic contacts. (B–D) Quantitative analyses of the SYP relative density (B), the number of SYP+ terminals per motor neuron per section (C), and the number of SYP+ terminals per dendrite per section (D). n = 6 mice/group. Error bars: mean ± SD. *p<0.05, **p<0.01, ***p<0.001, One-way ANOVA, Kruskal–Wallis post-hoc. Scale bars, 50 μm; Box scale bars, 20 μm. Abbreviations: AAV, adeno-associated virus (serotype 2); CTB, cholera toxin B subunit; DAPI, DAPI dihydrochloride; GFP, green fluorescent protein; NS, no significance; NT-3, neurotrophin-3; SCI, spinal cord injury; SD, standard error of the mean; SYP, synaptophysin.

DOI: https://doi.org/10.7554/eLife.39016.015

The following figure supplement is available for figure 8:

**Figure supplement 1.** Quantitative assessments of BDA+terminals surrounding ChAT+lumbar MNs.

DOI: https://doi.org/10.7554/eLife.39016.016

post-SCI. The rotarod data suggest that AAV-NT-3 significantly improved coordination or fatigue resistance. The TreadScan results suggest that AAV-NT-3 significantly restored pre-SCI gait. In addition to the behavioral improvements, tcMMEP recordings show that AAV-NT-3 treatment resulted in an increase in the number of early responses, an increase in amplitudes, and a decrease in latencies

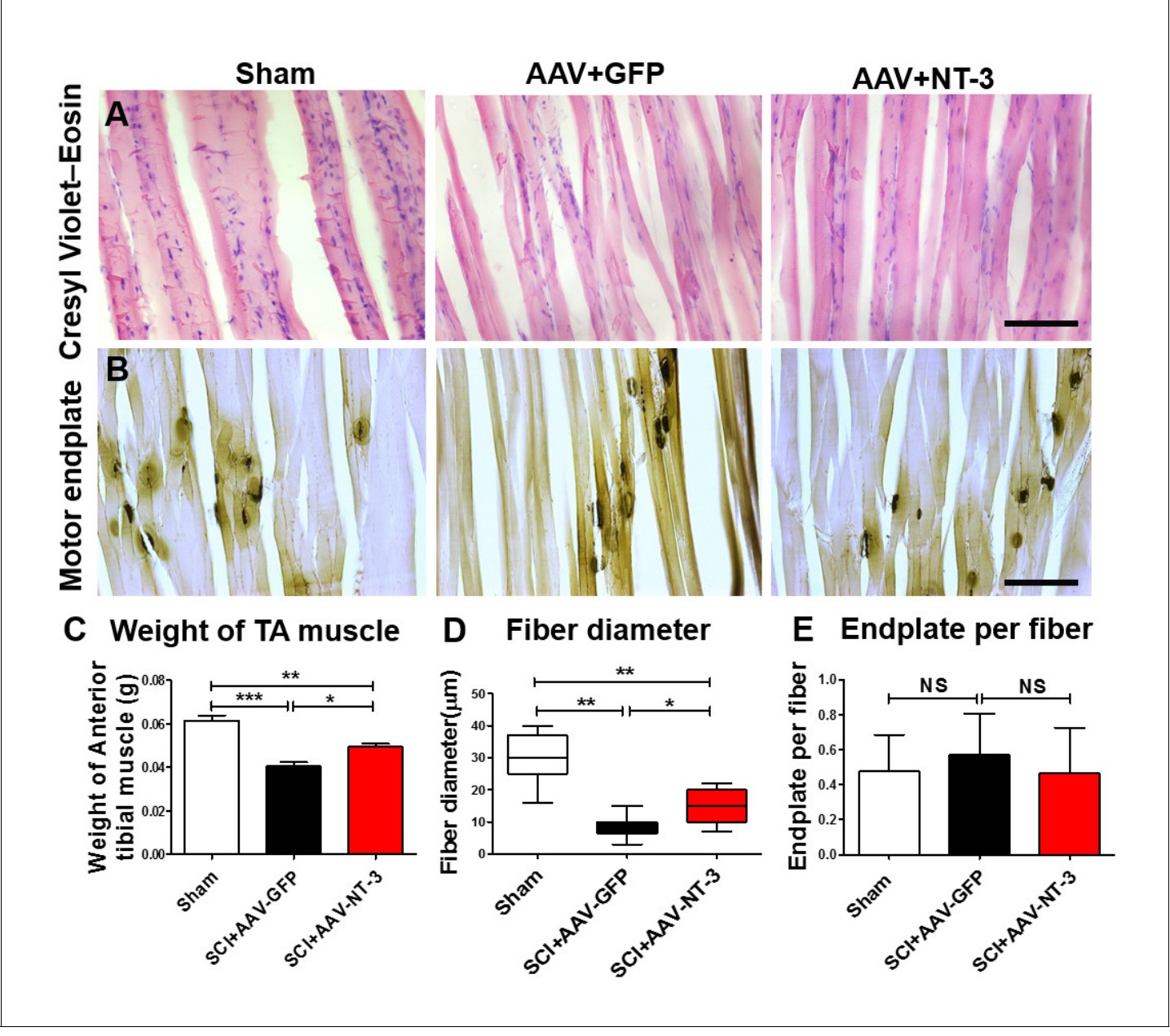

**Figure 9.** AAV-NT-3 increased muscle size and fiber diameter. (A and B) Cresyl Violet–Eosin (A) and acetylcholinesterase (B) staining in Sham, SCI + AAV GFP and SCI + AAV-NT-3 groups. (C – E) Quantitative analyses of TA muscle weight (C), fiber diameter (D) and endplate per fiber (E) (n = 6 mice/group). Error bars, mean ± SD. *p<0.05, **p<0.01, One-way ANOVA, Kruskal–Wallis post-hoc. Abbreviations: AAV, adeno-associated virus (serotype 2); GFP, green fluorescent protein; NS, no significance; NT-3, neurotrophin-3; SCI, spinal cord injury; SD, standard error of the mean; TA, tibialis anterior.

DOI: https://doi.org/10.7554/eLife.39016.017

The following figure supplement is available for figure 9:

**Figure supplement 1.** Histological comparison between AAV-GFP and AAV-NT-3 groups at 5 weeks post-SCI.

DOI: https://doi.org/10.7554/eLife.39016.018

in early and late responses at 5 weeks post-injury. These functional improvements correlated well with the improvement of several histological parameters of MNs, and attenuation of muscle atrophy.

## Retrogradely transported AAV-NT-3 prevented dendritic atrophy

After SCI, surviving MNs caudal to the lesion responded to injury with marked dendritic retraction (*Byers et al., 2012*). Changes in dendritic morphological patterns have drawn a lot of interest, particularly because dendritic morphology is a critical determinant of neuronal function. We confirmed that spinal MNs caudal to the lesion site underwent marked reduction in their dendritic arbor, especially in the dorsomedial portion of the ventral horn where both reticulospinal and propriospinal projections terminate (*Jones and Yang, 1985*; *Martin et al., 1985*), and disruption of which resulted in hindlimb motor deficits (*Loy et al., 2002*). Therefore, the dendritic atrophy of MNs caudal to a SCI we observed may reflect deafferentation resulting from the loss of descending motor and propriospinal tracts. A significant finding of this study is that the dendritic atrophy correlates well with behavioral deficits, suggesting that dendritic atrophy is an important contributor to movement deficits.

A previous study reported that growing dendrites analogously require anterograde NT-3 from their presynaptic partners during competitive dendrite growth (*Joo et al., 2014*). In our study, we found that AAV-NT-3 treatment prevented pronounced dendritic atrophy in lumbar MNs, indicating a protective role of NT-3 after SCI. The mechanism of NT-3-mediated protection of motoneuron dendritic atrophy may involve NT-3's retrograde transport to and releasing from motoneurons.

Motoneuron counts and somal size of TA MNs were not significantly affected by SCI, confirming that TA motoneurons caudal to the injury site remain morphologically intact and are not directly traumatized by the injury. Moreover, histological analysis showed no change in lesion area, volume, or myelin sparing in response to AAV-NT-3 treatment. A possible explanation of the lack of histological changes at the lesion site is that the T10 contusion level is rostral to the lumbar MNs and the retrogradely transported NT-3 did not reach the site of injury.

## Retrogradely transported AAV-NT-3 enhanced descending axonal projection to and synaptic connection with lumbar motoneurons

We have previously reported that contusive SCI results in the concomitant loss of descending pathways (*Liu et al., 2011*). Following SCI, CST axons fail to regenerate through and beyond the lesion site (*Sivasankaran et al., 2004*; *Liu et al., 2008*). The CST function, however, could be mediated by its projection to other brainstem and spinal pathways that spared beyond the lesion site, which could provide a relay transmission for supraspinal motor commands. We found that after T10 contusion, the disrupted dPST axons extended around the contusion cavity through spared ventrolateral white matter. Furthermore, the overexpression of NT-3 in lumbar MNs enhanced projection of dPST axons to these neurons. These data indicate that NT-3 is capable of promoting dPST axons to make new synaptic connections with lumbar MNs or ameliorate the SCI-induced loss of motor neuron inputs. To dissect between the two possibilities, a time course study showing the loss and recovery of descending inputs or synaptic boutons would be required.

Alterations of descending serotonergic and dopaminergic pathways have been correlated with sensory, motor, and autonomic disorders (*Hains et al., 2002*; *Lai et al., 2013*). After SCI, most regenerating fibers have been shown to be serotonergic (*Barritt et al., 2006*; *Alilain et al., 2011*). These fibers have multiple functions in providing excitatory input to the ventral horn MNs and controlling the motor rhythm of hind limbs (*Barbeau and Rossignol, 1991*; *Alvarez et al., 1998*). Correlation between recovery of locomotor function after SCI and the presence of 5-HT-positive fibers caudal to the lesion has been described previously (*Erschbamer et al., 2007*; *Mountney et al., 2010*). In this study, we found that AAV-NT-3 treatment enhanced serotonergic descending innervation of lumbar MNs which may form an anatomic basis for improved locomotor recovery.

Consistent with enhanced serotonergic projection, we found the increased projection of TH-IR descending dopaminergic axons to lumbar MNs following AAV-NT-3 treatment, demonstrating a role of descending dopaminergic projection in lumbar MN plasticity. A previous study demonstrated that NT-3 is localized to specific ventral mesencephalic regions containing dopaminergic cell bodies and that treatment with NT-3 resulted in dose-dependent increases in NT-3 expression in the number of TH-positive neurons, with similarly increased dopamine uptake activity and dopamine content (*Hyman et al., 1994*). Also, NT-3 has a broad effect on the survival, biosynthesis, and neuroprotection of dopaminergic neurons via autocrine or paracrine mechanisms (*Hyman et al., 1994*; *Seroogy et al., 1994*). It has been shown that descending serotonergic, dopaminergic, and

noradrenergic neurons express TrkB and TrkC (*King et al., 1999*; *Loudes et al., 1999*) and can be stimulated to regenerate with NT-3 treatment following SCI (*Bregman et al., 1997*).

It is possible that NT-3 displayed direct growth-stimulating and chemo-attractive effects on descending axons reaching the MNs after being retrogradely transported to the lumbar MNs. Indeed, NT-3 has been shown to induce chemotropic regeneration of axons after SCI (*Alto et al., 2009*; *Hollis et al., 2010*) and enhance axonal plasticity (*Schnell et al., 1994*). Furthermore, MNs and dendrites may locally secrete NT-3 to attract axons with the appropriate TrkC and/or p75 receptors, as shown in MNs, propriospinal neurons and interneurons of various sizes (*Merlio et al., 1992*; *Copray and Kernell, 2000*; *Gajewska-Woźniak et al., 2013*). Previous studies have shown that NT-3 could promote axon growth in multiple locomotor-related pathways including the CST, descending serotonergic and proprioceptive pathways (*Grill et al., 1997*; *Zhou et al., 2003*; *Gajewska-Woźniak et al., 2013*; *Lai et al., 2013*; *Weishaupt et al., 2014*).

A possible key step towards functional restoration in SCI is that AAV-NT-3 is ameliorating the SCI-induced loss of motor neuron inputs. One previous report demonstrated that AAV-NT-3 induced the strengthening of connections of ventral MNs and establishing novel functional synaptic connections from descending axons to the dorsomedial interneurons after thoracic contusion (*Hunanyan et al., 2013*). In our case, we found that AAV-NT-3 increased synaptic-like contacts on lumbar MNs and dendrites, suggesting that overexpression of NT-3 on MNs enhances synaptic input on lumbar MNs, potentially contributing to improved functional outcomes.

## AAV-NT-3 treatment prevented atrophy of hindlimb muscles

In the present study, AAV-NT-3 treatment also increased TA muscle fiber diameter and prevented muscle atrophy. Motor endplate densities, however, appeared unaffected. The protection of muscle fibers could be a result of enhanced motoneuron activity, which could be caused by multiple actions of the NT-3 including the prevention of dendritic retraction, and enhanced neural circuitry on MNs. Alternatively, NT-3 could be anterogradely transported to the terminals of MNs innervating the TA muscle to exert its protective effect directly on the muscle. Preventing muscle atrophy is an important component to preserve functional limb use after SCI.

In conclusion, we have demonstrated that retrogradely transported AAV-NT-3 prevented dendrite atrophy, promoted descending serotonergic, dopaminergic and propriospinal axonal innervation and synaptic formation on lumbar MNs, attenuated muscle atrophy, and, consequently, enhanced recovery of physiological and behavioral functions. These findings collectively suggest that retrograde transport of AAV-NT-3 may represent a promising therapeutic strategy for treating SCI.

# Materials and methods

All procedures were approved by the Institutional Animal Care and Use Committee of Indiana University School of Medicine (IACUC # 11011) and Institutional Biosafety Committee (IBC #1556), and were strictly following the National Institutes of Health (NIH) Guide on humane care and the use of laboratory animals.

## Viral vector preparation

An AAV-GFP virus containing both AAV serotype 2 Capsid and expressing GFP was prepared. GFP expression under the control of cytomegalovirus immediate-early promoter (CMV) (AAV-GFP, $1.0 \times 10^{13}$ viral particles/ml; Vector Biolabs, Philadelphia, USA) was used as a control. Human NT-3 subcloned into an AAV vector cassette under the control of the CMV and containing a polyadenylation signal from human B-globin gene was prepared by the Smith Lab (Temple University, Philadelphia, USA; $1.0 \times 10^{12}$ viral particles/ml).

## Primary spinal cord neuronal culture

Primary spinal cord neurons were obtained from Sprague-Dawley (SD) rat embryonic day 15 (E15) pups according to a previously established protocol (*Jiang et al., 2006*). In brief, E15 rat spinal cords were isolated and placed in L15 medium (*Wu et al., 2017*). Meninges were carefully removed and spinal cords were cut into small pieces, dissociated by incubation in 0.05% trypsin/EDTA 15 min at 37°C and triturated every 5 min. The dissociated cells were washed with and triturated in 10% heat-inactivated fetal bovine serum (FBS), 5% heat-inactivate horse serum (HS), 2 mM glutamine-DMEM

(all from Gibco, Grand Island, NY) and cultured in 10 cm plates for 30 min at 37°C to eliminate glial cells and fibroblasts. Neurons were plated on poly-L-lysine coated 48 well plates and incubated in a humidified atmosphere containing 5% $CO_2$ at 37°C with 10% FBS +5% HS+2 mM glutamine DMEM (all from Gibco). After 16 hr, medium was replaced with serum free neurobasal medium with 2% B27 (Gibco), 1% N2 (Gibco) and 2 mM glutamine (Gibco). On day 3, 5 μM cytosine-β-D-arabinofurano-side (Sigma-Aldrich, Saint Louis, MO) was added for 24 hr to inhibit glia cell proliferation. Using this culture protocol, a purity of >89% spinal cord neurons was obtained by 7 days *in vitro* (DIV). All experiments were performed between 7 – 10 DIV.

## AAV virus infection

The amount of transducing units of the vector stocks was determined by infecting spinal cord neuron cells (Abdellatif et al., 2006). The number of cells that were transduced with AAV-GFP was determined under UV radiation using a Zeiss microscope co-culture over 2 days. Five microscopic fields were randomly chosen, and positive cells were counted and multiplied by the dilution factor and 3769 (the number of microscopic fields needed to cover the 6 cm dish when using a 20 × objective). Titers were all in the range of $1 \times 10^{10}$ GM/ml.

## Dynamic neurite outgrowth assay

Spinal cord neurons were seeded at $2 \times 10^5$ per well in 48-well plates infected with AAV-GFP and AAV-NT-3 for 5 days. The dynamic outgrowth of neurites was recorded by an IncuCyte ZOOM Kinetic Imaging System (Essen BioScience, Ann Arbor, MI) every 4 hr for 5 days after virus treatment. Three views from each well were recorded and data were analyzed by an IncuCyte ZOOM software (Stewart et al., 2015) (n = 24 wells/group).

## NT-3 enzyme-linked immunosorbent assay (ELISA) *in vitro*

To determine the amount of NT-3 produced by AAV vector-transduced cells, spinal cord neurons were seeded into 48-well plates ($1.25 \times 10^5$ cells/well) for 5 days. On the 5th and 9th day, cells were infected with recombinant AAV-GFP (final concentration $6.7 \times 10^8$ viral particle/ml) and AAV-NT-3 ($1.3 \times 10^9$ viral particle/ml) at a multiplicity of infection (150 μl/well). The supernatant from AAV–GFP or -NT-3 virus treated cells was collected and quickly frozen in a dry ice/ethanol bath and kept at −80°C. The amount of secreted NT-3 protein was determined by ELISA (Human NT-3 DuoSet, R and D Systems, Inc., Minneapolis, MN) according to the manufacturer's protocol. The above experiments were repeated 3 times.

## Animals

A total of 66 adult, 18 – 22 g C57BL/6 mice (12 weeks, male n = 33 and female n = 33) were purchased from Jackson Laboratories (Bar Harbor, ME). All surgical interventions, treatments and postoperative animal care were performed in accordance with the Guide for the Care and Use of Laboratory Animals (National Research Council) and the Guidelines set forth by the Institutional Animal Care and Use Committee of the Indiana University School of Medicine.

**Table 1.** Summary of experimental design and animal groups

| Groups | Treatments | | Assessments | | | | |
|---|---|---|---|---|---|---|---|
| | T10 contusion | Sciatic nerve | HRP | BDA | CTB | Behavior | Total |
| Sham | - | PBS | n = 4 | (n = 4) | n = 10 | (n = 10) | n = 14 |
| SCI + AAV-GFP | + | Demyelination +AAV-GFP | n = 4 | (n = 4) | n = 13 | (n = 10) | n = 17 |
| SCI + AAV-NT-3 | + | Demyelination +AAV-NT-3 | n = 4 | (n = 4) | n = 13 | (n = 10) | n = 17 |

Note: HRP and BDA groups share the same animals. CTB and Behavior groups share the same animals. Abbreviations: AAV, adeno-associated virus, serotype 2; CTB, cholera toxin B; HRP, horseradish peroxidase; BDA, biotinylated dextran amine.
DOI: https://doi.org/10.7554/eLife.39016.019

In the first set of experiments, mice were randomly divided into three groups. Group 1: Sham (n = 14); Group 2: contusive SCI + AAV GFP (n = 17); Group 3: contusive SCI + AAV-NT-3 (n = 17) (*Figure 1A*; *Table 1*).

In the second set of experiments, animals were used to test the method of AAV-NT-3 retrograde transport after its injection into focal transiently demyelinated sciatic nerves. In this study, 18 adult mice randomly divided into two groups. Group 1: transient demyelination followed by AAV-GFP vector injection (n = 9); Group 2: transient demyelination followed by AAV-NT-3 vector (AAV-GFP and AAV-NT-3 were mixed, 1:1 ratio) injection (n = 9) (*Table 2*).

## Contusive SCI, transient focal demyelination, and viral injection

For contusive SCI, mice were anesthetized with a xylazine (10 mg/kg) and ketamine (100 mg/kg) mixture by intraperitoneal injection. A contusive SCI was then performed at the T10 vertebral level using the Louisville Injury System Apparatus (LISA) (Louisville, KY). The LISA impactor utilizes a laser sensor to measure the velocity and displacement of an injury obtained via a pneumatically driven impactor (*Zhang et al., 2008*). Mice in the SCI + AAV GFP and SCI + AAV-NT-3 groups received a 0.5 mm displacement contusion at a velocity of 1.0 m/s. Sham animals received laminectomy only. The overlying musculature was closed using suture and the skin was closed using wound clips. The animals were treated with Marcaine (Henry Schein, Melville, NY) at the incision site. A force/displacement graph was used to monitor impact consistency.

For transient demyelination, mice were immobilized in the prone position immediately after a contusive SCI. Sciatic nerves were exposed in the mid-thigh. Lysolecithin (Fischer Scientifics, Pittsburg, PA; 0.5 μl in 1% phosphate buffered saline (PBS)) was injected bilaterally into the sciatic nerves via a 32-gauge Hamilton syringe to produce demyelination (*Zhang et al., 2010*) (*Figure 1C*). The PBS injection group was used as a non-demyelination control. The needle was left in the nerve for 2 min following the injection to prevent leakage. After injection, muscles and skin were closed in layers. After surgery, animals were placed on a heating pad until full recovery from anesthesia was observed. Pain was monitored throughout the procedure and animals were treated with intraperitoneal (ip.) injection of buprenorphine (0.1 mg/kg, subcutaneous (sq.)) upon detection of discomfort signs. In addition, manual bladder expression was performed 2 – 3 times daily for 2 – 3 weeks until a voiding reflex returned and animals could leak urine.

Viral injections were performed at 5 days after demyelination. In this procedure, both sciatic nerves were re-exposed and injected with 1 μl of either AAV-GFP or AAV-NT-3 ($1.0 \times 10^{12}$ pfu/μl) using a 10 μl Hamilton syringe (Cole-Parmer, Vernon Hills, IL). We injected a small volume of high concentrations of viruses into the sciatic nerve because small volumes cause almost no damage to the injected nerves (*Zhang et al., 2010*). All injections were performed at a rate of 1 μl/min and the needle was held in place for an additional 2 min following the injection.

## Behavioral assessments
### Basso Mouse Scale

Basso Mouse Scale (BMS) locomotor test was performed weekly up to 5 weeks post-SCI (*Figure 1A*) by two observers lacking knowledge of the experimental groups according to a method published previously (*Basso et al., 2006*; *Liu et al., 2014b*). Briefly, mice were placed in an open field (diameter: 42 inch) and observed for 4 min by two trained observers. The scores were on a scale of 0–9 (0,

**Table 2.** Transient demyelination animal groups

| Groups | Treatments | | Assessments (time after demyelination) | | | | |
|---|---|---|---|---|---|---|---|
| | Virus | Sciatic nerve | 3 days IF | 5 weeks IF | 5 weeks ELISA | Behavior | Total |
| AAV-GFP | AAV-GFP | Transient demyelination | n = 3 | n = 3 | n = 3 | (n = 6) | n = 9 |
| AAV-NT-3 | AAV-GFP + AAV-NT-3(1:1) | Transient demyelination | n = 3 | n = 3 | n = 3 | (n = 6) | n = 9 |

Note: Behavior assessments used the same animals of the 5 week groups (IF and ELISA). Abbreviations: AAV, adeno-associated virus, serotype 2; ELISA, enzyme-linked immunosorbent assay; GFP, green fluorescent protein; IF, immunofluorescence; NT-3, neurotrophin-3.
DOI: https://doi.org/10.7554/eLife.39016.020

complete hind limb paralysis; 9, normal locomotion), which is based on hind limb movements made in an open field including hind limb joint movement, weight support, plantar stepping, coordination, paw position, and trunk and tail control.

## Rotarod

The rotarod performance test is a performance test based on a rotating rod with forced motor activity being applied. In this study, the rotarod test was performed on the 3rd and 5th week post-SCI (*Figure 1*) to assess balance and ability to coordinate stepping. Animals were placed on a single-lane rotarod according to our existing protocol (*Liu et al., 2013*) for a total of three trials per session. The rotarod was set for constant acceleration from 3 to 30 rpm over 300 s and animals were scored on seconds to fall. Each trial was scored individually and averaged for a final score per session.

## TreadScan analysis

TreadScan (CleverSys Inc., Reston, VA) is an unbiased device used for gait analysis of animals. All mice were allowed to walk on the motor-driven treadmill belt at a speed of 11 cm/s for a period of 20 s as described previously (*Beare et al., 2009*). The digital data (footprints and body movement) were analyzed by a TreadScan software (CleverSys Inc., Reston, VA), and each of the parameters was compared between ipsilateral and contralateral sides and between the treatment and control groups.

## Transcranial magnetic motor-evoked potentials

Nerve conduction was assessed by transcranial magnetic motor-evoked potentials (tcMMEP), an *in vivo* electrophysiological measure of motor pathway function which was previously described (*Liu et al., 2007*). The tcMMEP responses were elicited by the activation of subcortical structures with an electromagnetic coil placed over the cranium. Action potentials descend in the ventral spinal cord and synapse onto motoneuron pools which can be detected and recorded as output signals from both gastrocnemius muscles.

## Anterograde and retrograde tracings

### Anterograde tracing with biotinylated-dextran amine

On the 28th day after SCI, bilateral and stereotaxical injections of biotinylated-dextran amine (BDA; MW 10,000, 10%, 1 µl/site; Molecular Probes, Eugene, OR) were made into the intermediate gray matter of the T7-8 cord segments at distances of 3 – 6 mm rostral to the contusion site (for BDA, one injection/site/mm longitudinal distances) according to previously published work (*Deng et al., 2013*).

BDA staining was performed according to earlier reports (*Fouad et al., 2001*). Slides were dried in an incubator at 38°C for 1 hr and washed twice for 10 min in 50 mM Tris-buffered saline (TBS), pH 7.4, followed by two 45 min washes with TBS containing 0.5% Triton X-100. Afterwards, the slides were incubated overnight with an avidin–biotin–peroxidase complex in TBS with Triton (ABC Elite, Vector Laboratories, Burlingame, CA) according to the manufacturer's protocol. Subsequently, a DAB reaction was performed using the Vector DAB kit (SK4100, Vector Laboratories). The reaction was monitored and stopped by extensive washing in water.

### Retrograde tracing with cholera toxin B: injection into the sciatic nerve

For CTB retrograde tracing, a 2% solution of Alexa Fluor-labeled (594) cholera toxin B (CTB) (Invitrogen, Carlsbad, CA) in PBS was injected bilaterally into the sciatic nerves of the animals at 28 days following SCI. First, a small incision was made in the skin along the posterior thigh to expose the gluteus muscle. The muscle was separated to expose the sciatic nerve which was subsequently injected with 2 – 3 µl of 2% CTB using a Hamilton syringe with a 30 gauge needle according to an existing protocol (*Hirakawa and Kawata, 1992*). For this injection, the sciatic nerve was not crushed. Mice were monitored for 7 days following injection of the neural tracer.

## Retrograde tracing with cholera toxin B: injection into the tibialis anterior

Horseradish peroxidase conjugated to the cholera toxin B subunit (BHRP) was also used for the retrograde tracing. On the 33$^{rd}$ day after SCI, animals were re-anesthetized and the tibialis anterior (TA) muscle was exposed and injected with BHRP (0.5 µl, 0.2%; List Biological Laboratories, Inc., Campbell, CA). BHRP labeling permits population-level quantitative analysis of motoneuron somal and dendritic morphologies. Twenty-eight hours after the BHRP injection, a period that ensures optimal labeling of motoneurons, animals were euthanized and perfused intracardially with saline followed by cold fixative (1% paraformaldehyde/1.25% glutaraldehyde) (*Byers et al., 2012*). The number of animals that received anterograde or retrograde tracings is summarized in *Table 1*.

### Dendritic morphology

#### Dendritic length measurement

For each animal, dendritic lengths taken in a single representative set of alternate sections were measured under dark field illumination according to our existing protocol (*Liu et al., 2014a*). Beginning with the first section in which BHRP-labeled fibers were present, labeling through the entire rostrocaudal extent of the TA motoneuron dendritic field was assessed in every third section (480 µm apart) using a three dimensional, computer-based morphometry system (Neurolucida; MBF Bioscience, Williston, VT) at a final magnification of 250×. Average dendritic length per labeled motoneuron was estimated by summing the measured dendritic lengths of the series of sections, multiplying by three to correct for sampling, then dividing by the total number of labeled motoneurons in that series. This method does not attempt to assess the actual total dendritic length of labeled motoneurons but has been shown to be a sensitive and reliable indicator of changes in dendritic morphology during normal development, after changes in dendritic interactions and afferent input, and after injury (*Byers et al., 2012*).

#### Dendritic distribution measurement

To assess potential dendritic redistributions across groups, for each animal the composite dendritic arbor created by the length analysis was divided using a set of axes oriented radially around the center of the collectively labeled somata. These axes divided the spinal cord into 12 bins of 30° each. The portion of each animal's dendritic arbor per labeled MN contained within each bin was then determined. This method provided a sensitive measure of dendritic redistribution in response to changes in dendritic interactions and afferent input (*Goldstein et al., 1993*; *Byers et al., 2012*; *Liu et al., 2014a*; *Wang et al., 2015*).

### NT-3 ELISA *in vivo*

The lumbar spinal cords collected from each group were dissected and quickly frozen in a dry ice/ethanol bath and kept at −80°C. The amount of secreted NT-3 protein was determined by an ELISA (Human NT-3 DuoSet, R and D Systems, Inc, Minneapolis, MN) according to our existing protocol (*Abdellatif et al., 2006*). The experiment was repeated >3 times.

### Histological assessments

Animals were perfused, and the cord segments containing the injury epicenter were dissected out, embedded, and sectioned at 20 µm according to our previous protocols (*Liu et al., 2011*). A set of serial spinal cord cross sections were stained for myelin with Luxol fast blue and counterstained with Cresyl Violet–Eosin. Lesioned areas were outlined and quantified using an Olympus BX60 microscope equipped with a Neurolucida system (MBF Bioscience, Williston, VT). Areas of the lesion at the injury epicenter were quantified using NIH imaging software (NIH) and serial sections centered at the lesion epicenter were traced in ImageJ (NIH) for volume calculation, according to a previous publication (*Liu et al., 2014b*).

### Immunofluorescence staining

The biological activity of the AAV-NT-3 was determined using a spinal cord dendrite outgrowth assay. The cultured spinal cord neurons were grown for 9 days at 37°C, 5% $CO_2$ followed by fixation in 4% paraformaldehyde for 10 min. Dendrite outgrowth and NT-3 expression in these cultures was

evaluated by staining with an anti-β-tubulin antibody (1:200, Sigma, St. Louis, MO) and an NT-3 antibody (1:10,000. Santa Cruz Biotec., Santa Cruz, CA).

For *in vivo* immunofluorescence staining, animals were transcardially perfused with ice-cold PBS followed by 4% paraformaldehyde at 5 weeks after SCI. Dissected spinal cords were postfixed in 4% paraformaldehyde overnight and then transferred to a 30% sucrose solution before cryostat sectioning of the selected cord segments including the injured T10 segment and lumbar enlargement (cross sections at 25 μm), TA muscle (longitudinal sections at 30 μm) and L3-5 dorsal root ganglia (cross sections at 25 μm; embedded in 10% gelatin). Sciatic nerves were dissected at 3 days and 5 weeks after SCI and were embedded in 10% gelatin before sectioning at 6 μm in longitudinal sections.

Immunofluorescent staining method was performed according to our existing protocol (*Liu et al., 2006*; *Liu et al., 2014b*). Briefly, slides were warmed for 15 min on a slide warmer at 37°C and then rinsed three times in 0.01 M PBS for a duration of 10 min each. All sections were first blocked with 2% normal bovine serum for 1 hr, followed by overnight incubation at 4°C with combinations of primary antibodies (listed in *Table 3*). After several washes, immunoreactive sites were revealed by using species-specific secondary antibodies (listed in *Table 3*). After staining, the sections were rinsed three times in PBS and cover slipped with antifading aqueous mounting medium (Biomeda Corp., Foster City, CA) and viewed with a confocal microscope (Zeiss, Germany). Pixel intensity was measured on images taken on a standard fluorescent microscope (Zeiss, Germany) with a uniform exposure setting and analyzed using ImageJ (NIH).

To analyze the synaptic surroundings of the motoneurons, the motoneurons were automatically selected and a constant threshold was used to segment and obtain an estimated average density for each labeling. Immunoreactivity was evaluated in a perimeter of 5 μm width surrounding the soma. This 5 μm-width perimeter covered the synaptic area surrounding the motoneurons and dendrites, limiting the overlapping with synapses of neighboring motoneurons (*Arbat-Plana et al., 2015*). For each animal, 10 to 15 motoneurons from each pool and each side were analyzed.

## Muscle fiber and motor endplate density

The TA muscles were removed immediately after perfusion and weighed. Muscles were then postfixed overnight and transferred to sucrose phosphate buffer (10% w/v, pH 7.4). Muscles were then rinsed in distilled water, blocked into proximal and distal segments, and flash-frozen in 2-methylbutane. Muscle segments were sectioned longitudinally on a cryostat at −20°C and thaw-mounted onto glass slides. Muscle fiber diameters were assessed after staining with Cresyl Violet–Eosin stain (*Liu et al., 2014a*). Motor endplate densities were assessed after staining for acetylcholinesterase using the Roots-Karnovsky method (*Byers et al., 2012*). Muscle fiber diameters and motor endplate densities were measured under a bright-field illumination using Stereo Investigator (MBF Bioscience, Williston, VT). The number of motor endplates per muscle fiber was estimated by counting the number of muscle fibers and endplates in a grid (1 mm ×1 mm) randomly placed on the muscle section (one sample field per section, five muscle sections per animal). Fiber and endplate areas within each animal were then averaged for statistical analysis.

## Data acquisition and statistical analysis

To blind researchers to the treatment groups during behavioral assessments, surgeries, and electrophysiological recordings, we established a standard practice of coding all animals with numbers that were randomized and not reflective of treatment groups. Treatment information was separated from the coded numbers immediately following treatment assignment and was not present during these procedures. One animal in the SCI + AAV GFP group and one animal in the SCI + AAV-NT-3 group were removed from the study and euthanized due to bladder infection and complication.

Data are presented as mean ± SD. All statistical values were calculated using GraphPad Prism 5.0 software (GraphPad Software, Inc., San Diego, CA). Sample sizes were initially determined using statistical software to calculate the minimum total required a number of animals or assays. All reported groups were above the minimum calculated sample size.

All statistical analysis was performed by two-tailed Student's t-tests, one-way ANOVA, and Tukey's posthoc test between all groups in each experiment. The early response rate of tcMMEP was performed with $\chi^2$ test and BMS analysis was performed with repeated measures two-way ANOVA.

**Table 3.** Primary and secondary antibodies used in this study

| Antigen | Host type | Working dilution | Manufacturer | Catalog# | Research resource identifiers (RRIDs) |
|---|---|---|---|---|---|
| Glial fibrillary acidic protein (GFAP) | Mouse Monoclonal IgG | 1:1000 | Invitrogen | G9269 | AB_477035 |
| β−3-Tubulin | Mouse Monoclonal IgG | 1:200 | Sigma, St. Louis, USA | T5293 | AB_477580 |
| Choline acetyltransferase (ChAT) | Goat Monoclonal IgG | 1:200 | Sigma, St. Louis, USA | ABIN350213 | AB_10781260 |
| CS-56 (CSPG) | Mouse Monoclonal IgM | 1:200 | Sigma, St. Louis, USA | C8035 | AB_476879 |
| ED-1 (CD-68) | Rabbit Monoclonal IgG | 1:200 | AbD Serotec, USA | MCA341R | AB_2291300 |
| Green fluorescent protein (GFP) | Chicken Monoclonal IgG | 1:1000 | Chemicon, USA | AB16901 | AB_11212200 |
| 5-Hydroxytryptamine (5-HT) | Mouse Monoclonal IgG | 1:200 | Abcam, USA | ab85615 | AB_10696528 |
| NeuN | Mouse Monoclonal IgG | 1:200 | Chemicon, USA | MAB377 | AB_2298772 |
| Neurofilament 200 (NF) | Rabbit Monoclonal IgG | 1:200 | Sigma, St. Louis, USA | N4142 | AB_477272 |
| Synaptophysin (SYP) | Mouse Monoclonal IgG | 1:1000 | Millipore/Life Technologies, USA | MAB5258 | AB_2313839 |
| SMI-31 | Mouse Monoclonal IgG | 1:200 | Covance, USA | SMI-31R-100 | AB_10122491 |
| S-100 | Mouse Monoclonal IgG | 1:200 | Sigma, St. Louis, USA | HPA015768 | AB_1856538 |
| Tyrosine Hydroxylase (TH) | Mouse Monoclonal IgG | 1:200 | Sigma, St. Louis, USA | T9573 | AB_261823 |
| Neurotrophin-3 (NT-3) | Rabbit Polyclonal IgG | 1:1000 | Santa Cruz Biotechnology, Santa Cruz, USA | sc-80250 | AB_1126615 |
| SecondaryAntibodies | | | | | |
| Alexa Fluor 488 (green) | Goat anti Chicken | 1:1000 | Invitrogen, USA | A11039 | AB_142924 |
| ExtrAvidin−FITC buffered aqueous solution | | 1:200 | Sigma, St. Louis, USA | E2761 | AB_2492295 |
| ExtrAvidin−TRITC buffered aqueous solution | | 1:200 | Sigma, St. Louis, USA | E3011 | AB_2492295 |
| FITC conjungated anti rabbit secondary antibody | Goat Polyclonal IgG | 1:200 | Sigma, St. Louis, USA | F0382 | AB_259384 |
| FITC conjungated anti mouse secondary antibody | Goat Polyclonal IgG | 1:200 | Sigma, St. Louis, USA | F5262 | AB_259638 |
| TRITC conjungated anti rabbit secondary antibody | Goat Polyclonal IgG | 1:200 | Sigma, St. Louis, USA | T6778 | AB_261740 |
| TRITC conjungated anti mouse secondary antibody | Goat Polyclonal IgG | 1:200 | Sigma, St. Louis, USA | T2402 | AB_261618 |
| CY5 conjungated anti mouse secondary antibody | Goat Polyclonal IgG | 1:200 | Invitrogen, USA | A10524 | AB_2534033 |
| Hoechst 33342 | | 1:200 | Sigma, St. Louis, USA | 4082S | AB_10626776 |

DOI: https://doi.org/10.7554/eLife.39016.021

Regression analysis was performed in behavioral and dendrite length examinations for each animal. Regression values (*r*) were identified by comparing two individual variables. The Pearson correlation coefficients were identified for parametric data sets.

## Acknowledgements

This work was supported in part by NIH R01 NS059622, R01 NS103481, R01 NS100531; Merit Review Award I01 BX002356, I01 BX003705 from the U.S. Department of Veterans Affairs, DOD CDMRP W81XWH-12-1-0562, Craig H Neilsen Foundation 296749 and 382267, Indiana State Department of Health 019919 (XMX), and Science Fund Project of Natural Science Foundation of China 81870977 (YW).

## Additional information

### Funding

| Funder | Grant reference number | Author |
| --- | --- | --- |
| National Natural Science Foundation of China | Science Fund Project 81870977 | Ying Wang |
| Craig H. Neilsen Foundation | 382267 | Ling-Xiao Deng |
| National Institutes of Health | R01 NS103481 | Xiao-Ming Xu |
| U.S. Department of Veterans Affairs | I01 RX002356-01 | Xiao-Ming Xu |
| Craig H. Neilsen Foundation | 296749 | Xiao-Ming Xu |
| Indiana State Department of Health | 019919 | Xiao-Ming Xu |
| National Institutes of Health | R01 NS100531 | Xiao-Ming Xu |
| U.S. Department of Veterans Affairs | I01 BX003705-01A1 | Xiao-Ming Xu |
| National Institutes of Health | NS059622 | Xiao-Ming Xu |
| U.S. Department of Defense | CDMRP W81XWH-12-1-0562 | Xiao-Ming Xu |

The funders had no role in study design, data collection and interpretation, or the decision to submit the work for publication.

### Author contributions

Ying Wang, Data curation, Formal analysis, Methodology, Project administration; Wei Wu, Data curation, Formal analysis, Validation, Methodology; Xiangbing Wu, Yan Sun, Yi P Zhang, Ling-Xiao Deng, Melissa Jane Walker, Nai-Kui Liu, Heqiao Dai, Formal analysis, Validation, Methodology; Wenrui Qu, Data curation, Validation, Methodology; Chen Chen, Formal analysis, Validation, Methodology, Writing—review and editing; Qi Han, Data curation, Formal analysis, Methodology; Lisa BE Shields, Resources, Writing—review and editing; Christopher B Shields, Conceptualization, Resources, Supervision, Methodology; Dale R Sengelaub, Formal analysis, Supervision, Methodology, Writing—review and editing; Kathryn J Jones, Conceptualization, Supervision, Funding acquisition, Writing—review and editing; George M Smith, Conceptualization, Resources, Data curation, Funding acquisition, Methodology, Writing—review and editing; Xiao-Ming Xu, Conceptualization, Resources, Supervision, Funding acquisition, Methodology, Writing—original draft, Project administration, Writing—review and editing

### Author ORCIDs

Chen Chen (iD) http://orcid.org/0000-0001-5039-3066
Lisa BE Shields (iD) http://orcid.org/0000-0002-1526-4063
Xiao-Ming Xu (iD) http://orcid.org/0000-0002-7229-0081

### Ethics

Animal experimentation: All surgical interventions, treatments, and postoperative animal care were performed following the Guide for the Care and Use of Laboratory Animals (National Research

Council) and the Guidelines set forth by the Institutional Animal Care and Use Committee of the Indiana University School of Medicine.

## Decision letter and Author response
Decision letter https://doi.org/10.7554/eLife.39016.026
Author response https://doi.org/10.7554/eLife.39016.027

# Additional files

## Supplementary files
• Transparent reporting form
DOI: https://doi.org/10.7554/eLife.39016.022

## Data availability
All data generated or analyzed during this study are included in the manuscript and supporting files.

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
