## [Decision Letter]

Thank you for submitting your article "Remodeling of lumbar motor circuitry remote to a thoracic spinal cord injury promotes locomotor recovery" for consideration by *eLife*. Your article has been reviewed by three peer reviewers, and the evaluation has been overseen by a Reviewing Editor and Gary Westbrook as the Senior Editor. The following individuals involved in review of your submission have agreed to reveal their identity: Mary Bartlett Bunge (Reviewer **#**1) and Joe Springer (Reviewer **#**2).

The reviewers have discussed the reviews with one another and the Reviewing Editor has drafted this decision to help you prepare a revised submission.

Summary:

This manuscript reports that delivering NT-3 to the lumbar motor neurons via appropriate viral injection into the sciatic nerve aids in recovery after a spinal cord contusion injury at T10. The paper is clearly written and the results are well shown in the figures. Quite a number of techniques have been used. Of particular note are the complementary approaches demonstrating that enhanced recovery of locomotor function is strongly correlated with enhanced sprouting of several descending pathways, improved tcMMEP activity and inhibition of hindlimb muscle atrophy. This is clearly a novel contribution to the spinal cord literature.

Major issues:

These issues should be addressed with either additional data or discussion in the text.

1) There is clear evidence that certain descending supra- and intraspinal pathways are sensitive to the actions of NT-3. This reviewer was surprised to read that NT-3 overexpression enhanced sprouting of the descending dopaminergic pathways. It would be helpful if the authors cited evidence supporting the actions of NT-3 on this dopamine system. Do these neurons contain TrkC? If not, what is the mechanism by which NT-3 is affecting this neuronal population?

2) The authors report a 20-fold increase in NT-3 levels in animals receiving AAV-NT-3 targeted delivery. This is a significant amount that is being secreted over an extended period of time. Did the investigators observe any aberrant effects not specific to motor neuron function? What other neuronal or glial systems might be affected by NT-3 overexpression? It is reported that DRG sensory neurons overexpress NT-3 and some of these are sensitive to the actions of this neurotrophin. However, it was not clear (or I might have missed it) whether the AAV-NT-3 treatment resulted in any changes in sensory function. Some comments would be helpful.

3) From the data presented, it is not clear as to whether NT-3 is ameliorating the SCI-induced loss of motor neuron inputs, or promoting the formation of new connections as suggested by the authors. The former mechanism seems much more likely than the latter. The assertion throughout that NT-3 is supporting a reconstitution of structures lost to injury is not supported by the evidence herein. In the absence of further experiments, the assertions that NT-3 is enhancing structural changes must be edited. A time course showing the loss and recovery of descending inputs or synaptic boutons would be required to demonstrate that targeted NT-3 expression can promote the formation of new structures and synapses.

4) There is considerable discussion about the effects on descending propriospinal axons, however injection of the anterograde tracer BDA is likely to label axons of passage as well as propriospinal neurons. What is the density of staining in Figure 7D-F relative to? Is BDA labeling normalized to the number of labeled, surviving axons at the level of injury? What molecular weight BDA was used in the studies? Is there a difference in BDA labeling around the injury site with AAV-NT-3 transduction? Subsection “Retrogradely transported AAV-NT-3 enhanced descending axonal projection to and synaptic connection with lumbar motoneurons”, second paragraph, the reduced attenuation of TH, 5-HT, or BDA-labeled fibers does not demonstrate that any of these specific axonal populations play a role in the recovery of function.

---

## [Author Response]

Major issues:These issues should be addressed with either additional data or discussion in the text.1) There is clear evidence that certain descending supra- and intraspinal pathways are sensitive to the actions of NT-3. This reviewer was surprised to read that NT-3 overexpression enhanced sprouting of the descending dopaminergic pathways. It would be helpful if the authors cited evidence supporting the actions of NT-3 on this dopamine system. Do these neurons contain TrkC? If not, what is the mechanism by which NT-3 is affecting this neuronal population?

We appreciate the reviewer’s comment. In the revised Discussion, we have added a paragraph, cited below, to discuss the action of NT-3 on the dopamine system:

“A previous study demonstrated that NT-3 is localized to specific ventral mesencephalic regions containing dopaminergic cell bodies and that treatment with NT-3 resulted in dose-dependent increases in NT-3 expression in the number of TH-positive neurons, with similarly increased dopamine uptake activity and dopamine content (Hyman et al., 1994). […] It has been shown that descending serotonergic, dopaminergic, and noradrenergic neurons express TrkB and TrkC (King et al., 1999, Loudes et al., 1999) and can be stimulated to regenerate with NT-3 treatment following SCI (Bregman et al., 1997).”

2) The authors report a 20-fold increase in NT-3 levels in animals receiving AAV-NT-3 targeted delivery. This is a significant amount that is being secreted over an extended period of time. Did the investigators observe any aberrant effects not specific to motor neuron function? What other neuronal or glial systems might be affected by NT-3 overexpression? It is reported that DRG sensory neurons overexpress NT-3 and some of these are sensitive to the actions of this neurotrophin. However, it was not clear (or I might have missed it) whether the AAV-NT-3 treatment resulted in any changes in sensory function. Some comments would be helpful.

The reviewer has raised an important issue: whether AAV-NT-3 might cause some aberrant effects. The following are our responses to this issue.

i) The previous study showed that AAV was taken up by peripheral axons and transported in a retrograde manner to motoneurons and DRG neurons (Zhang et al., 2010). To assess whether NT-3 overexpression in a retrograde manner affected neurons other than MNs or glial cells, we measured the expression of a general neuronal marker NeuN and an astrocyte marker GFAP in sections of the lumbar spinal cord using immunostaining. Results showed that AAV_2_-NT-3 treatment had no effect on the neuronal number (*t* = 1.026, *df* = 10, *P* = 0.322, ns) and astrocytic integrated optical density (IOD) (*t* = 0.141, *df* = 10, *P* = 0.889, ns) between the SCI + AAV-GFP and SCI + AAV-NT-3 groups in the L2-L5 spinal cord segments (see Author response image 1). Due to the space limitation, we did not include this data in our original manuscript, and will not include it in the revision. Moreover, our results indicate that a thoracic contusive SCI did not cause lumbar MN loss (see Figure 6—figure supplement 1). In general, we didn’t observe any aberrant effects of AAV-NT-3 on other neurons and astrocytes.

ii) Our results also showed AAV-GFP expression in DRG neurons and sensory axons after sciatic nerve injection of AAV-GFP (Figure 3). In the revised manuscript, we added: “Our results were in agreement with a previous report using a similar approach (Zhang et al., 2010).”

iii) We found that, within the L2-L5 DRGs, NT-3 immunoreactivity (IR) in the AAV-NT-3 injection group was significantly higher than that in the AAV-GFP injection group. It is likely that DRG neurons are sensitive to the actions of NT-3. Further studies need to be conducted to determine the effect of AAV-NT-3 treatment on sensory function following SCI.

**Author response image 1. respfig1:** Effect of NT-3 on neuronal number and astrocytic integrated optical density (IOD) in the lumbar spinal cord. (**A**) Representative images of neurons (NeuN-immunoreactive, IR) or astrocytes (GFAP-IR) in SCI + AAV-GFP and SCI + AAV-NT-3 treated animals in the lumbar spinal cord segments. (B-C) No statistical significant differences were found in the NeuN-IR neuronal number (**B**) and GFAP-IR IOD (**C**) between the two groups. Student’s t tests.

3) From the data presented, it is not clear as to whether NT-3 is ameliorating the SCI-induced loss of motor neuron inputs, or promoting the formation of new connections as suggested by the authors. The former mechanism seems much more likely than the latter. The assertion throughout that NT-3 is supporting a reconstitution of structures lost to injury is not supported by the evidence herein. In the absence of further experiments, the assertions that NT-3 is enhancing structural changes must be edited. A time course showing the loss and recovery of descending inputs or synaptic boutons would be required to demonstrate that targeted NT-3 expression can promote the formation of new structures and synapses.

We totally agree with the reviewer’s point of view. Although we propose that NT-3 overexpression by MNs and their dendrites promote the formation of new synaptic connections, we cannot rule out the possibility that NT-3 may ameliorate the SCI-induced loss of motor neuron inputs. We have incorporated the latter possibility in our Discussion. We agree that a time course study showing the loss and recovery of descending inputs or synaptic boutons would be required to demonstrate that targeted NT-3 expression can promote the formation of new structures and synapses. In the revised Discussion, we wrote: “These data indicate that NT-3 is capable of promoting dPST axons to make new synaptic connections with lumbar MNs or ameliorate the SCI-induced loss of motor neuron inputs. To dissect between the two possibilities, a time course study showing the loss and recovery of descending inputs or synaptic boutons would be required.”

4) There is considerable discussion about the effects on descending propriospinal axons, however injection of the anterograde tracer BDA is likely to label axons of passage as well as propriospinal neurons. What is the density of staining in Figure 7D-F relative to? Is BDA labeling normalized to the number of labeled, surviving axons at the level of injury? What molecular weight BDA was used in the studies? Is there a difference in BDA labeling around the injury site with AAV-NT-3 transduction? Subsection “Retrogradely transported AAV-NT-3 enhanced descending axonal projection to and synaptic connection with lumbar motoneurons”, second paragraph, the reduced attenuation of TH, 5-HT, or BDA-labeled fibers does not demonstrate that any of these specific axonal populations play a role in the recovery of function.

i) Since we injected a small amount (1 µL) of BDA into the intermediate gray matter of the spinal cord, it would be unlikely to label the axons of passage (in the white matter).

ii) The density we used was the integrated optical density (IOD)/mm^2^ in figure 7D-F. In the revision, we wrote: “(D-F) Quantitative analyses of the TH^+^ relative integrated optical density (IOD)/mm^2^ (D), 5-HT^+^ IOD (E), and BDA^+^ IOD (F).”

iii) No, we did not normalize the BDA labeling. Rather, we compare the treatment groups with the Sham group as a control.

iv) The molecular weight of BDA that we used is 10,000. We have indicated it in the Materials and methods section.

v) No, there was no difference in BDA labeling around the injury site between SCI + AAV-GFP and SCI + AAV-NT-3 groups (*t* = 0.232, *df* = 10, *P* = 0.818, ns). Due to the space limitation, we will not include this data in our revised manuscript.

vi) Yes, the data in this paragraph does not indicate that these specific axonal populations play a role in the recovery of function, so “recovery of function” has been deleted.

**Author response image 2. respfig2:** BDA relative density in SCI + AAV-GFP and SCI + AAV-NT-3 groups (n = 6/group). mean ± SD, Student’s t tests.